# Ad Auctions for LLMs via Retrieval Augmented Generation

**MohammadTaghi Hajiaghayi**
University of Maryland
hajiagha@umd.edu

**Sébastien Lahaie**
Google Research
slahaie@google.com

**Keivan Rezaei**
University of Maryland
krezaei@umd.edu

**Suho Shin**
University of Maryland
suhoshin@umd.edu

## Abstract

In the field of computational advertising, the integration of ads into the outputs of large language models (LLMs) presents an opportunity to support these services without compromising content integrity. This paper introduces novel auction mechanisms for ad allocation and pricing within the textual outputs of LLMs, leveraging retrieval-augmented generation (RAG). We propose a *segment auction* where an ad is probabilistically retrieved for each discourse segment (paragraph, section, or entire output) according to its bid and relevance, following the RAG framework, and priced according to competing bids. We show that our auction maximizes logarithmic social welfare, a new notion of welfare that balances allocation efficiency and fairness, and we characterize the associated incentive-compatible pricing rule. These results are extended to multi-ad allocation per segment. An empirical evaluation validates the feasibility and effectiveness of our approach over several ad auction scenarios, and exhibits inherent tradeoffs in metrics as we allow the LLM more flexibility to allocate ads.

## 1 Introduction

Large language models (LLMs) [4, 1, 39] have recently gained widespread attention, serving various functions including question answering, content generation, translation, and code completion [31, 13, 44, 24]. The emergence of AI-driven assistant models like ChatGPT, Gemini, and Claude has influenced how individuals interact with these technologies, as they increasingly use them to streamline and enhance their work.

While LLMs provide a fresh way to engage with information, the most advanced models are costly to operate [27]. To date, online advertising has been one of the most successful business models of the digital economy. Ads support a wide variety of online content and services, ranging from search engines, online publishers, to video content and more. However, LLM services today predominantly follow a subscription model [32]. A natural question to ask in this context is whether advertising could support LLMs to alleviate serving costs and charges to users, and what format advertising on LLMs might take.

In this paper, we develop auctions that allocate online ads within the output of LLMs using the framework of *retrieval augmented generation* (RAG) [23]. RAG is one of the most popular techniques to integrate factual information into the output of LLMs. When a query is submitted by a user, RAG first retrieves the top-$k$ most relevant documents for the query from a database, and then conditions on these documents to generate the LLM's output, significantly enhancing the reliability of the generated

38th Conference on Neural Information Processing Systems (NeurIPS 2024).

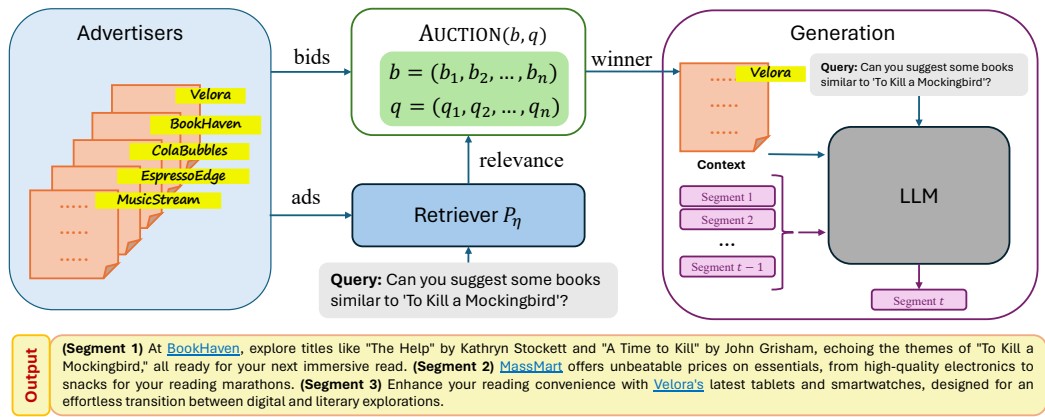

Figure 1: Segment auction architecture for LLMs via RAG.

content [14]. The RAG framework naturally lends itself to ad allocation, by retrieving relevant ads from a database rather than documents. The ads can then be incorporated into the output of the LLM via a variety of methods, most directly via prompt engineering.

In the auction design literature, a typical approach is to start with a desirable social choice function, and derive an allocation rule that maximizes this function [34]. Here, we take the RAG allocation as given, and investigate which social choice function it corresponds to and what associated pricing rules give it good incentive properties. In particular, we are interested in pricing rules that are *individually rational*, such that there is always an incentive to participate, and ideally *incentive compatible*, such that advertisers are motivated to report their true willingness to pay for their ad to be shown [15, 35, 21].

**Our Contributions.** We introduce the concept of a *segment auction*, in which ads are allocated for each discourse segment, which could be a sentence, paragraph, or the entire LLM output. The architecture of a segment auction is depicted in Figure 1. Given a user query to the LLM, relevant ads together with bids are retrieved from a database. The retriever forwards the bids to the auction, along with click probabilities (aligned with retrieval probabilities). The auction implements a *randomized* allocation rule based on these inputs, following the RAG framework, and the LLM bases its output on the winning ad. The auction can be run repeatedly for each segment, or it can compute several winners for multiple segments at once.

We first consider the case where a single ad is allocated per segment, and later study the generalization to multiple ads per segment. For single-ad allocation per segment, we show that the RAG-based allocation rule is optimal with respect to a new notion of logarithmic social welfare (LSW), a type of welfare function that balances economic efficiency and fairness. This balance is highly desirable for LLM outputs, which need to be satisfactory to users while potentially generating ad revenue. We show how the randomized RAG allocation rule can be obtained as a randomization over deterministic truthful auctions, which directly leads to a truthful implementation of RAG allocation. We confirm that the expected payment under this scheme matches the pricing rule that obtains from Myerson's lemma [28]. For the general setting of multi-ad allocation per segment, we again provide an auction obtained as a randomization over deterministic truthful auctions. The main device for single- and multi-ad allocation is to perturb the bids with random additive offsets, drawing on ideas from discrete choice methods [41].

We validate the feasibility and effectiveness of our approach via experiments using publicly-available LLM APIs. We compare single- and multi-allocation segment auctions against each other, and against two naive baselines that do not consider relevance scores, or do not use the LLM to integrate ads (simply appending them to the output instead). Our key finding is that whereas repeated single-ad segment auctions generate higher revenue, a multi-allocation auction leads to higher output quality, as measured by the cosine similarity between embeddings of the output omitting ads, and the output conditioned on ads. We corroborate these results with a qualitative analysis of outputs from single- and multi-allocation segment auctions. We conclude by discussing remaining practical challenges in implementing segment auctions.

**Related Work.** The question of using auctions to influence LLM output has been examined by a few very recent papers. Feizi et al. [12] present a high-level framework for LLM-based advertising and discuss key requirements such as privacy, latency, and reliability. Duetting et al. [10] propose a *token auction* to aggregate the outputs of several distinct LLMs, weighted by bids. The motivation is that the LLMs can be provided by different competing advertisers. Under this approach every single token is the result of an auction to choose the source LLM, whereas in our work advertisers bid more traditionally to be placed in some segments of the output (e.g., the first paragraph).

Soumalias et al. [37] provide an auction framework for agents (e.g., advertisers) to steer LLM output according to their preferences, which can be represented by their own LLM or directly via a reward function. Their approach is closely linked to reinforcement learning with human feedback (RLHF), rather than RAG. Candidate outputs are generated by conditioning on context from each advertiser, and one of these outputs is sampled according to aggregate reward across agents. Our approach does not necessarily require generating output candidates for each ad, although this can increase allocative efficiency, as we discuss. Along similar lines, Sun et al. [38] investigate how to design mechanisms that incentivize truthful reporting of preferences when fine-tuning LLMs to cater to multiple user groups.

Dubey et al. [9] introduce a factorized framework for LLM ad auctions where an auction module allocates prominence (representing relative importance) to each ad based on their bids and predicted click-through rates. Although prominence could in practice map to ad selection probabilities under RAG, their paper focuses on the concrete application of generating ad summaries within the LLM output. The auction module's prominence allocation serves as a guide to control the output of the LLM module, ensuring that ads with higher prominence receive longer mentions and more user attention. The framework is designed to be incentive compatible and to maximize social welfare by ensuring high-quality summaries and efficient allocation of ad space.

Since the introduction of RAG by Lewis et al. [23], the technique has gained widespread interest in academia and industry. The retrieval component is crucial for our work, with key sub-components including the embedding model for efficient document similarity search [7, 36, 45] and query optimization to incorporate contextual information [47, 8]. For more details on RAG, see the recent survey by Gao et al. [14] or tutorial by Asai et al. [3].

## 2 Preliminaries

We now formally define the model behind LLM auctions for ad allocation via RAG. Consider a set of ads indexed by $[n] = \{1, 2, \ldots, n\}$, where each ad $i$ is provided by an advertiser. (We will refer to the ad and advertiser interchangeably for simplicity.) We often write $a_i$ to denote the $i$-th ad for clarity. Each advertiser $a_i$ has a private valuation $v_i$ for its ad to be clicked. Each advertiser, possibly strategically, submits a bid $b_i$ to be shown in the output in the LLM and to maximize its own payoff, which will be defined shortly. We write $x$ to denote a user query (a prompt), $y$ to denote the output generated from the LLM, and $y^{(i:j)}$ to denote the sub-sequence of $y$ from $i$-th token to $j$-th token. We use boldface to denote a vector, *e.g.,* $\mathbf{b} = (b_1, b_2, \ldots, b_n)$ is the vector of bids.

When a user enters a query $x$ into the LLM, an auction mechanism selects an ad $a_i$ to advertise, and generates an output $y_i$ that includes a mention of ad $a_i$ along with a hyperlink. By design, once the ad $a_i$ to advertise is decided, the generation of $y_i$ is independent of the auction and bids $\mathbf{b}$.

**Retrieval augmented generation.** To build some intuition on how an LLM auction would operate under the RAG framework, we first recap how the original formulation of RAG proceeds given a set of documents [23]. Given a query $x$, suppose there exists a set of documents $\{z_1, z_2, \ldots, z_n\}$ that can be used to inform the output of the LLM. Under RAG, the output follows the generative model:

$$P(y|x) = \sum_{i \in \text{top-}k(P_\eta(\cdot|x))} P_\eta(z_i|x) P_\theta(y|x, z_i), \tag{1}$$

where the summation is over the top-$k$ documents with highest $P_\eta(z_i|x)$, retrieved via a technique like maximum inner-product search [19, 18]. The $\eta$ and $\theta$ here refer to the parameters of the retrieval and generator components, either of which can be fine-tuned for overall RAG performance. In practice, RAG is often implemented simply by including information from the selected document $z_i$ into the prompt for generating $y$. The model in (1) refers to a variant called RAG-sequence: a

single document is selected probabilistically, and is then used to inform the entire output sequence. Lewis et al. [23] also introduce a variant called RAG-token, where a separate document is selected and considered for each token generation. The flexibility of RAG to consider different contextual inputs at separate stages of output generation is one of the features that make it particularly suitable for integrating ads. We emphasize that document retrieval in (1) is *randomized*; RAG does not just deterministically retrieve the highest-scoring document. This probabilistic integration helps the model to leverage information from multiple sources and enhances the robustness and accuracy of the generated responses.

**Auction design.** To define an auction under the RAG framework, we must specify ad selection probabilities given submitted bids, along with prices for ads that are selected. We assume that RAG provides baseline ad selection probabilities $P_\eta(a_i|x)$ given a query $x$; these are the probabilities that should hold if all ads bid equally, so that there is no reason to prefer any ad based on the bids. We make the following important assumption that the retrieval component is calibrated to the expected clicks an ad $a_i$ would receive under query $x$, *i.e.,* its click-through rate (ctr$_i$).[1]

**Assumption 2.1** (Calibrated Retriever). Let $C_i$ be the binary event that ad $i$ is clicked. We assume that $\text{ctr}_i := \mathbf{E}[C_i|x] = \sum_{y_i} \mathbf{E}[C_i|x, y_i] P(y_i|x) \propto P_\eta(a_i|x) =: q_i$ where $y_i$ is the output generated from the query $x$ augmented with the ad $a_i$.

For ad retrieval purposes, the system could allocate according to $\mathbf{E}[C_i|x, y_i]$ after observing the output $y_i$ generated by conditioning on each ad $i$. Our model accommodates such generalization at the cost of additional query complexity; we elaborate on this in Appendix A and B. However, from the perspective of an advertiser, all strategizing happens before the output is generated, taking randomness in ad retrieval and output generation into account.

The quantity $q_i$ can be seen as an indirect measure of the *relevance* of $a_i$ to query $x$. Throughout we assume that advertisers are charged *per-click*. However, in mechanism design it is more standard to work with expected prices *per-impression*.[2] The per-click price $p_i$ and per-impression price $p_i$ for ad $i$ are related by $\tilde{p}_i = \text{ctr}_i \cdot p_i$, by Assumption (2.1). Given this assumption, we emphasize that our auction would not directly require information on the ctr$_i$, but only requires the retrieved relevance $q_i$ to run the entire mechanism and generate the output, due to inherent normalization in the auction allocation rule. We will elaborate on this shortly.

An auction defines an allocation rule $\mathbf{x}(\mathbf{b})$, where $x_i(\mathbf{b})$ is the selection probability of ad $i$ under the given bids (which implicitly also depends on the baseline selection probabilities).[3] Note that under RAG, the allocation rule is naturally randomized. The auction also defines a payment rule $\mathbf{p}(\mathbf{b})$, where $p_i(\mathbf{b})$ is the expected *per-impression* payment of ad $i$. Given ad $i$'s private value-per-click $v_i$, its ex-ante utility (namely, its expected utility before ad selection and clicks are realized) is defined as $\tilde{u}_i(\mathbf{b}) = \text{ctr}_i \cdot v_i x_i(\mathbf{b}) - \tilde{p}_i(\mathbf{b}) = \text{ctr}_i(v_i x_i(\mathbf{b}) - p_i(\mathbf{b})) \propto q_i(v_i x_i(\mathbf{b}) - p_i(\mathbf{b}))$. We write per-click utility as $u_i(\mathbf{b}) = v_i x_i(\mathbf{b}) - p_i(\mathbf{b})$. An auction is *dominant-strategy incentive-compatible* (DSIC) if it is optimal for ad $a_i$ to report its true value to the auction, holding the other bids fixed: $u_i(v_i, \mathbf{b}_{-i}) \geq u_i(b_i, \mathbf{b}_{-i})$ for all possible bids $b_i$ and competing bids $\mathbf{b}_{-i}$. An auction is *individually rational* if no advertiser is worse off by participating in the auction: $u_i(v_i, \mathbf{b}_{-i}) \geq 0$ for all $\mathbf{b}_{-i}$.

# 3 Single allocation segment auction

Following the RAG framework, we introduce a *segment auction* to retrieve and allocate ads during the LLM's process of output generation. A discourse *segment* is an abstraction of a series of tokens that will be the minimal unit of generation for the LLM auction. For example, the segment could be a single token, sentence, paragraph, or even an entire document. The segment size can be enforced at a low-level by truncating tokens; otherwise, prompt engineering can be quite effective at limiting LLM output to a specific number of sentences or paragraphs [12, 40].

---

[1]The retrieval component can be calibrated using a number of standard methods, such as Platt scaling or Bayesian binning [33, 29]. We also refer to [25] and [16] for descriptions of actual click-through rate estimation systems at Google and Microsoft using calibration via isotonic regression.

[2]In advertising terms, an "impression" refers to the instance when an advertisement is viewed once by a user, or displayed once on a webpage.

[3]With a slight abuse of notation, we use $x$ without any subscript to refer to the user query, whereas $\mathbf{x}$ or $x_i$ denotes the allocation probabilities, following conventions in RAG and auction theory.



**Single allocation segment auction**

1. Collect $\mathbf{q}$ and $\mathbf{b}$.
2. Draw $\varepsilon_i \sim \text{Gumbel}(0,1)$ for each $i \in [n]$ independently.
3. Compute the score $s_i = q_i b_i e^{\varepsilon_i}$.
4. Select the winner $w = \text{argmax}_{i \in [n]} s_i$.
5. Find the second highest $\ell = \text{argmax}_{i \in [n] \setminus \{w\}} s_i$.
6. Find the smallest bid $z$ for $w$ such that $s_w \geq s_\ell$, which is $z = q_\ell b_\ell e^{\varepsilon_\ell} / q_w e^{\varepsilon_w}$.
7. Charge $z$ to ad $a_w$ *per click*.



Figure 2: Single allocation segment auction

We first focus on the scenario in which a single ad is incorporated into each segment; we consider the generalization to multiple ads per segment in Section 3.2. Let $T$ be the number of segments. Formally, we are interested in generating the $t$-th segment $y^{(t)}$, given the series of previous segments $y^{(1:t-1)}$. In generating each segment $y^{(t)}$, we have an opportunity to incorporate one of $k$ ads into the output. The RAG generative model is as follows.

$$P(y^{(1:T)}|x) = \prod_{t \in [T]} \sum_{i \in [n]} P_\eta(a_i|x, y^{(1:t-1)}; \mathbf{b}) \, P_\theta(y^{(t)}|x, y^{(1:t-1)}, a_i). \tag{2}$$

The probability $P_\eta(a_i|x, y^{(1:t-1)}; \mathbf{b})$ is an adjustment of $P_\eta(a_i|x, y^{(1:t-1)})$ according the advertisers' bids. We will focus on the following adjusted probability based on linear aggregation [10]:

$$\hat{q}_i^{(t)} = \frac{b_i \cdot q_i^{(t)}}{\left(\sum_{j \in [n]} b_j \cdot q_i^{(t)}\right)}. \tag{3}$$

Note that if all the bids are the same, this reduces to the baseline RAG output model.

We further impose the following assumption asserting that each segment is rich enough to capture click-through rate by itself, and importantly, the advertiser's utility is additive over each segment.

**Assumption 3.1** (Rich Segment). Let $C_i^{(t)}$ be the binary event that ad $a_i$ is clicked in the $t$-th segment. We assume that Assumption 2.1 holds segment-wise, so that $\text{ctr}_i^{(t)} := \mathbf{E}[C_i^{(t)}|x, y^{(1:t-1)}] \propto P_\eta(a_i|x, y^{(1:t-1)}) =: q_i^{(t)}$ for each $t \in [T]$. We also assume that utility decomposes additively across segments: $u_i = \sum_{t \in [T]} u_i^{(t)}(\mathbf{b})$.

Due to this assumption, one can observe that (3) is equivalent to $b_i \cdot \text{ctr}_i^{(t)} / (\sum_{j \in [n]} b_j \cdot \text{ctr}_i^{(t)})$. Thus, it suffices to only deal with the calibrated relevance instead of the actual click-through rates here.

Our segment auction is presented formally in Figure 2. The key idea is to first perturb each agent's score (i.e., bid-per-impression $q_i b_i$) using independent Gumbel random variables, and then run a standard second-price auction. The bid perturbation ensures that bidders win following adjusted probabilities (3), *e.g.*, see Lemma H.1. This kind of perturbation is known as the Gumbel max-trick, and is also a familiar idea in discrete choice methods in econometrics [17, 41]. After the segment auction determines the allocation and payment, the LLM outputs $y^{(t)}$ according to the generative model (2).[4]

### 3.1 Theoretical analysis

We now provide a theoretical analysis of the segment auction. In the mechanism design literature, the auctioneer is typically interested in (1) incentive-compatibility, so that truth telling is a dominant strategy, (2) individual rationality, so that no participant is ever worse off by by participating in the

---

[4]We focus on the segment auction with replacement, where the same ad can be selected multiple times across different segments. We also implement the segment auction without replacement in Section 4.

auction. We mainly consider the following *independent segment auction* such that the relevance is independent from the previous segments, *i.e.,* for every $t \in [T]$:

$$q_i^{(t)} = q_i \propto P_\eta(a_i|x). \tag{4}$$

A slightly more general model would be $q_i^{(t)} = \delta^{(t)} q_i$, where $\delta^{(t)}$ is a segment-wise factor that can capture a user's decreasing propensity to click as the ad is shown in later segments in the output.[5] Our results extend in a straightforward fashion to monotonically decreasing segment factors, so we omit them for simplicity.

Our first result is that for the class of independent segment auctions, segment auction maximizes a new notion of logarithmic social welfare assuming truthful bidding, and further satisfies desired properties. All the proofs can be found in Appendix H.

**Theorem 3.2.** *Given a query $x$, the segment auction is DSIC, IR, Pareto-efficient, and has the maximal logarithmic social welfare (henceforth LSW) among independent segment auctions, where LSW is defined by* [6]

$$\mathrm{LSW} = \prod_{t \in [T]} \mathrm{LSW}^{(t)} = \prod_{t \in [T]} \prod_{i \in [n]} (x_i^{(t)})^{v_i q_i}.$$

Recall that we write $x_i$ to denote a component of the allocation vector and $x$ to denote the user query (see footnote 3). We remark that even though LSW is defined over $q_i$, replacing it with $\mathrm{ctr}_i$ induces the same optimization problem due to the calibrated relevance assumptions.

Intuitively, if the mechanism sets $x_i^{(t)} = 0$ for some $i \in [n]$, then $\mathrm{LSW}^{(t)}$ becomes zero, implying that the mechanism should guarantee some positive probability of selection to every ad for every segment. Note that this is indeed a logarithmic analogue of the social welfare since $\log(\mathrm{LSW}^{(t)}) = \sum_{i \in [n]} v_i q_i \log x_i^{(t)}$. Investigating further properties of the proposed notion of logarithmic social welfare remains as an interesting open question.

**Theorem 3.3.** *The segment auction is a randomization over truthful auctions. For the $t$-th segment, its per-click payment rule takes the form*

$$\frac{w_{-i}}{q_i} \left( \ln \left( \frac{q_i b_i + w_{-i}}{w_{-i}} \right) - \frac{q_i b_i}{w_{-i} + q_i b_i} \right), \tag{5}$$

*where $w_{-i} = \sum_{j \neq i} q_j b_j$. Any truthful auction for RAG allocation rule (3) has per-click payment rule (5), up to an additive constant.*

The segment auction is truthful, as it is a randomization over truthful second-price auctions [26]. The fact that payment (5) is unique up to an additive constant follows from Myerson's lemma [28].

## 3.2 Multi-allocation segment auction

So far, we have focused on the setting in which the auction mechanism only advertises a single ad per segment. This approach, however, might be wasteful if the segment is long enough to adapt multiple ads, or if there are several ads that can be advertised naturally without compromising the segment's quality. In this section, we propose a multi-allocation segment auction that allocates multiple ads in a single segment. The main question here is how one can design the allocation and payment function to obtain a mechanism that exhibits several desired properties. To this end, we formally consider the auction procedure for the $t$-th segment. Assuming that we are interested in selecting $k$ ads for each segment, it proceeds as depicted in Figure 3. Given that the mechanism selects the set of winners $A^* \in \mathcal{A}_k$ where $\mathcal{A}_k = \{A \subseteq [n] : |A| = k\}$, we delegate the role of generating the output $y_{A^*}$ entirely to the LLM. For instance, we provide a single document that concatenates the descriptions of the ads in $A^*$ and query the LLM to generate the output conditioned on such a document.

The following theorem characterizes the allocation function for the multi-allocation segment auction.

---

[5] This is analogous to position effects in search advertising auctions [11, 43].

[6] This differs slightly from the well-known Nash social welfare but can be viewed as a version of weighted Nash social welfare with a certain structure. Since weighted Nash social welfare satisfies several fairness notions such as weighted proportional fairness and competitive equilibrium from equal income (CEEI), LSW also guarantees versions of these fairness criteria. Detailed discussion is provided in Appendix D.

> **Multi-allocation segment auction**
> 1. Collect $\mathbf{q}$ and $\mathbf{b}$.
> 2. Draw $\varepsilon_i \sim \text{Gumbel}(0, 1)$ for each $i \in [n]$ independently.
> 3. Compute the score $s_i = q_i b_i e^{\varepsilon_i}$.
> 4. Sort the bidders so that $s_{\sigma(1)} \geq s_{\sigma(2)} \geq \ldots \geq s_{\sigma(n)}$ for some permutation $\sigma$ over $[n]$.
> 5. Select the winners $A^* = \{\sigma(1), \ldots, \sigma(k)\}$.
> 6. For each winner $\sigma(i)$ for $i \in [k]$, find the smallest bid $z_i$ such that $s_{\sigma(i)} \geq s_{\sigma(k+1)}$, which is $z = q_{\sigma(k+1)} b_{\sigma(k+1)} e^{\varepsilon_{\sigma(k+1)}} / q_{\sigma(i)} e^{\varepsilon_{\sigma(i)}}$.
> 7. Charge $z_i$ to each winner $\sigma(i)$ per click.

Figure 3: Multi-allocation segment auction.

**Theorem 3.4.** *Let $\bar{S} = [n] \setminus S$. For each $S \in \mathcal{A}_k$, the probability that the set of ads $S$ is selected as the winners is*

$$\mathbb{P}(S \text{ wins}) = \sum_{T \subseteq S} (-1)^{|T|+1} \frac{\sum_{j \in T} q_j b_j}{\sum_{i \in \bar{S} \cup T} q_i b_i}.$$

Indeed, this strictly generalizes the single allocation segment auction since taking $S = \{i\}$, we get

$$\mathbb{P}(\{i\} \text{ wins}) = (-1)^{|\{i\}|+1} \frac{\sum_{j \in \{i\}} q_j b_j}{\sum_{j \in \bar{S} \cup \{i\}} q_j b_j} = \frac{q_i b_i}{\sum_{j \in N} q_j b_j},$$

which is the standard selection probability for the single-ad setting.

In fact, the described multi-allocation segment auction can be deemed as a special case of more general *combinatorial* segment auction (see Appendix E). Briefly speaking, in the combinatorial segment auction, we consider each set $A \in \mathcal{A}_k$ as a single entity to retrieve in RAG, and we assign a set-wise relevance metric $q_A$ to obtain the allocation probability of each set, which is further decomposed by the individual relevance $q_{A,i}$ of each ad $i \in A$.[7] One advantage of the combinatorial segment auction is that the individual relevance $q_{A,i}$ given the set $A$ can be more naturally connected with the advertiser utility, making it easier to calibrate with the actual click-through rate. However, this comes at the cost of larger computational complexity and query complexity with respect to the relevance and the LLM module, and requires to compute the individual relevance as well as the set-wise relevance.

## 4 Experiments

We validate our theoretical findings and provide insights on operating segment auctions in practice via numerical simulations. After determining the winning advertiser, we provide the ad to the LLM context and ask the model to generate an additional segment incorporating the ad, while continuing from the previous segments and advertising the selected ad. More details on the LLM and prompts used are in Appendix F.

**Setup.** We consider segment auctions where each segment is a single sentence, and the entire document consists of three segments. There are five types of auctions: (1) segment auction with replacement, allowing repeated selection of the same ads across segments, (2) segment auction without replacement, requiring different ads for each segment, (3) Naive I, which uses the same allocation and payment functions as the segment auction with replacement but concatenates selected ads' texts at the end of the output, without using the LLM to integrate them into the output, (4) Naive II, similar to single allocation segment auctions but disregarding relevance (see Pseudocode in Appendix F.2, Figure 5), and (5) multi-allocation auction, which treats the three sentences as one longer segment, allocating three ads to the entire output at once.

---

[7]We further characterize the VCG payment that makes the combinatorial segment auction DSIC, IR, and formally prove that it further maximizes a combinatorial version of the LSW. Details can be found in Appendix E.

**Relevance measure.** To compute the relevance measure, we use a model from the `sentence-transformers` library.[8] This model maps input sentences or paragraphs into an embedding space where semantically similar texts have higher cosine similarity, while unrelated texts have lower cosine similarity.

**Scenarios.** We consider three experimental scenarios, each involving a set of advertisers and their corresponding bids. However, due to space limitations we defer the analysis of two scenarios to the Appendix. We run each scenario 500 times (trials) and calculate the average of the metrics, which will be defined shortly. Throughout the experiments, the query is fixed to: *"Can you suggest some books similar to 'To Kill a Mockingbird'?"*.

**Auction outcomes.** We calculate the following outcome metrics for evaluation:

- Revenue $:= \sum_{t\in[T]} \sum_{i\in[n]} p_i^{(t)}$.
- Social welfare $:= \sum_{t\in[T]} \sum_{i\in[n]} v_i q_i^{(t)} x_i^{(t)}$. [9]
- Relevance $:= \sum_{t\in[T]} \sum_{i\in[n]} q_i^{(t)} x_i^{(t)}$.
- Minimum social welfare $:= \min_{i\in[n]} \sum_{j\in[500]} \tilde{u}_{i,j}$ where $\tilde{u}_{i,j}$ denotes the allocative utility $\sum_{t\in[T]} v_i q_i^{(t)} x_i^{(t)}$ of agent $i$ in trial $j$.

**Quality of output.** To capture the genuine quality of the generated output, inspired by Feizi et al. [12], we measure the *embedding similarity* between the original outputs in which no ads are advertised but rather purely generated from the LLM, and the modified output corresponding to each mechanism. This similarity is computed using cosine similarity and is normalized to lie in $[0, 1]$.

### 4.1 Results

**Auction outcomes.** The scenario we consider is shown on the left of Table 1 and includes four ads with a wide range of final allocation probabilities. Note that we do not include the results for Naive I here since its auction metrics are necessarily identical to the segment auction with replacement. As seen on the right of Table 1, social welfare varies significantly between mechanisms. The segment auction with replacement has the highest social welfare, followed by the one without replacement, due to large differences in allocative efficiency ($q_i v_i$) among advertisers. The segment auction without replacement still outperforms Naive II in terms of social welfare. However, the multi-allocation segment auction tends to have the lowest revenue as it charges lower payments to winners.

| Advertiser | Bid | $q_i$ | $x_i$ | Mechanism | Soc. Wel. | Revenue | Relevance | Min. Soc. Wel. |
|---|---|---|---|---|---|---|---|---|
| Velora | 3 | 0.36 | 0.22 | Seg w/ repl. | .660 (±.0091) | .371 (±.0070) | .688 (±.0082) | .185 |
| BookHaven | 3 | 0.87 | 0.54 | Seg w/o repl. | .521 (±.0025) | .333 (±.0060) | .565 (±.0021) | .294 |
| MassMart | 2 | 0.31 | 0.13 | Naive II | .508 (±.0085) | .379 (±.0065) | .552 (±.0076) | .329 |
| EspressoEdge | 2 | 0.26 | 0.11 | Multi alloc | .524 (±.0021) | .238 (±.0061) | .569 (±.0016) | .298 |

Table 1: Experiment setup (left), and the corresponding auction outcomes (right). Note that all metrics are normalized by dividing them by their maximum possible value.

Notably, the relevance of the segment auction with replacement far exceeds that without replacement, unlike the uniform scenario. This is due to the ad 'Bookhaven' with very large relevance (0.87). Naive II and the segment auction without replacement have similar overall relevance, likely because the number of ads (4) is small compared to the total number of slots (one for each of 3 segments). For minimum social welfare, the ordering is opposite to social welfare due to differences in allocation probabilities. Segment auction with replacement selects 'Bookhaven' repeatedly, while without replacement, different ads are chosen for different slots. Naive II has slightly larger minimum social welfare than the segment auction without replacement due to a more uniform selection procedure

---

[8]https://huggingface.co/sentence-transformers/multi-qa-MiniLM-L6-cos-v1

[9]We here use $q_i$ instead of the exact click-through rate. Given the calibrated relevance assumption, all the mechanisms' social welfare would be equivalently scaled, so all the discussions carry over with the exact click-through rate.

induced by similar bids in Table 1. Similar tendencies are observed in the other two scenarios, detailed in Appendix G.1.

**Output quality.** To further verify the effect of incorporating the set of ads in the context of the entire output, we implement the multi-allocation segment auction. For a fair comparison, in this auction, we define the segment to be the entire three sentences, and allocate $k = 3$ ads within it.

| Mechanism | 1st seg | 2nd seg | 3rd seg | $k = 1$ | $k = 2$ | $k = 3$ |
|---|---|---|---|---|---|---|
| Seg w/ repl. | .746 ($\pm$.0040) | .596 ($\pm$.0040) | .588 ($\pm$.0039) | .746 ($\pm$.0040) | .715 ($\pm$.0039) | .700 ($\pm$.0036) |
| Seg w/o repl. | .752 ($\pm$.0040) | .602 ($\pm$.0045) | .576 ($\pm$.0043) | .752 ($\pm$.0040) | .716 ($\pm$.0035) | .702 ($\pm$.0034) |
| Naive I | .743 ($\pm$.0043) | .555 ($\pm$.0033) | .551 ($\pm$.0035) | .743 ($\pm$.0043) | .740 ($\pm$.0044) | .671 ($\pm$.0032) |
| Naive II | .745 ($\pm$.0048) | .600 ($\pm$.0040) | .584 ($\pm$.0047) | .745 ($\pm$.0048) | .712 ($\pm$.0045) | .698 ($\pm$.0040) |
| Multi-alloc | - | - | - | - | - | .715 ($\pm$.0030) |

Table 2: The 2-4th columns represent the similarity of the individual segment to the original output, and the 5-7th columns represent the similarity of the first $k$ segments to the original output.

Interestingly, we indeed observe that the eventual output quality of the multi-allocation segment auction is the highest among every mechanism. This implies that if one gives flexibility of deciding which part to incorporate ads among the entire document, the LLM would do a much better job articulating the advertised output without compromising the output quality. In Appendix G.1, we observe that this phenomenon is amplified with more advertisers, demonstrating the superiority of the multi-allocation segment auction in terms of output quality.

**Qualitative analysis of single vs. multi-allocation.** We provide a qualitative comparison of the output quality between the single and multi-allocation segment auctions with 11 advertisers in Figure 4. In the multi-allocation segment auction, interestingly, we observe that an ad is not always advertised in each sentence. Indeed, both 'MassMart' and 'EspressoEdge' are advertised in the second sentence in a coherent manner (teal text), while the first sentence is devoted to fully addressing the user's query. In addition, the third sentence (teal text) is constructed in a cohesive way from the previous sentence.

This verifies our conjecture that the multi-allocation segment auction constructs more coherent output, as the LLM can optimize over the entire document to incorporate the selected ads. The LLM can decide a coherent ordering of ads within the document, unlike the single allocation segment auction, which forces one ad into each segment. For example, in the first segment of the single allocation auction, advertising 'BrainChips' is less cohesive (red text), while the second segment is more fluent (teal text), and the third segment becomes irrelevant (red text) again. Further qualitative analyses are provided in Appendix G.2.

---

**Single allocation**:

**(Segment 1)** If you appreciate the classic exploration of moral and ethical questions in "To Kill a Mockingbird," you might find "The Help" by Kathryn Stockett equally compelling, providing rich narratives driven by the kind of complex social issues that users of BrainChips technology tackle every day in their quest for innovative solutions and understanding. **(Segment 2)** Reflect on these intricate tales of humanity and justice with a good book in hand, and perhaps your next reading adventure can kick off with a delightful shopping trip to MassMart, where you can explore their extensive range of high-tech electronics and more, enhancing your reading experience with unbeatable value and convenience. **(Segment 3)** After securing your new reads, unwind with a visit to EspressoEdge, where the rich, meticulously crafted beverages offer the perfect accompaniment to dive into your literary journey, reinforcing a truly immersive experience with each sip.

**Multi-allocation**:

If you enjoyed the profound themes of racial justice and moral growth in "To Kill a Mockingbird," then I suggest checking out "The Help" by Kathryn Stockett and "Go Set a Watchman" by Harper Lee, which explores similar veins of social and ethical dilemmas. While you're picking up these intriguing reads at MassMart, where high-quality products meet unbeatable prices, perhaps consider enhancing your reading experience with a comforting cup of coffee from EspressoEdge, renowned for its exquisite blends perfect for literary afternoons. And for those who prefer digital reading, make sure your devices are powered by BrainChips processors, ensuring a smooth, efficient reading experience that keeps you immersed in the world of justice and personal integrity.

Figure 4: Outputs of single and multi-allocation segment auctions.

# 5 Conclusions

This paper considered the question of integrating ads into the output of LLMs, to offset serving costs and user subscription charges. Our approach is based on the popular RAG framework for incorporating factual information into LLM-generated content [23]. We introduced the concept of a *segment auction* where an auction is run to integrate single or multiple ads into each output segment (e.g., sentence, paragraph). We showed that our segment auction designs implement the RAG allocation rule while charging incentive compatible prices. We also showed that the single-ad segment auction maximizes logarithmic social welfare, which balances efficiency and fairness in the allocation. In our experimental evaluation, our key finding was that whereas repeated single-ad segment auctions have higher revenue, less-frequent multi-ad auctions lead to higher quality output, for the same number of ads. This uncovers an inherent trade-off between revenue and quality for operators of segment auctions.

We see several avenues for follow-up work. We note that while our work takes the perspective of charging ads to appear in LLM output, one could also take the reverse perspective where information sources need to be compensated for providing unique, factual information under the RAG framework. In that case a *reverse auction* would be run to obtain high-quality information at minimum cost. We expect the main design ideas presented here to carry through, though important practical details (e.g., per-impression vs. per-click pricing) would change. Another question is the integration of *reserve prices*, which can serve to both increase revenue and maintain output quality standards. Finally, based on our findings on the relative quality of single- vs. multi-ad segment auction output, it would be worthwhile to investigate more sophisticated approaches to RAG segment auctions like joint fine-tuning of the retriever and generator components.

# 6 Acknowledgements

This work is partially supported by DARPA QuICC, NSF AF:Small #2218678, NSF AF:Small #2114269, Army-Research Laboratory (ARL) #W911NF2410052, and MURI on Algorithms, Learning and Game Theory.

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

# A   Ex-post relevance

Recall that each allocation probability $q_i^{(t)}$ depends on the RAG probability $p_\eta(a_i|x)$, and we assume that the average CTR is proportional to this quantity.[10] Note further that this is obtained from a retriever component which typically uses the relevance between two documents to compute the similarity. Alternatively, one may consider incorporating the output into the context to derive the *ex-post relevance*, $P_\eta(a_i|x, y_i, y^{(1:t-1)})$, where $y_i$ the generated output when $a_i$ is selected.[11] We denote the previous notion of relevance which marginalizes over the output by *ex-ante* relevance. There can be several ways to compare such relevance, *e.g.,* concatenating the query $x$ and the output $y_i$ and computes its merged document's relevance to the ad document. This approach, however, comes at the cost of the additional computation complexity because one needs to precompute every possible $y_i$ for $i \in [n]$ to obtain the ex-post relevance. This will further be discussed shortly in the complexity paragraph.

Further, similar to Assumption 2.1, we may assume that our retrieval component to measure the ex-post relevance is calibrated to the ex-post CTR.

**Assumption A.1.** Let $C_i$ be the binary event that ad $i$ is clicked. We assume that $\tilde{q}_i^{(t)} = \mathbf{E}[C_i^{(t)}|x, y_i, y^{(1:t-1)}] \propto P_\eta(a_i|x, y_i, y^{(1:t-1)})$.

Then, assuming that the ex-post relevance is calibrated to the ex-post CTR, the segment auction with ex-post relevance ensures that allocation function always achieve better objective function.

**Proposition A.2.** *Given a query $x$ and for any objective function $f : \Delta_n^T$, the optimal allocation with ex-post relevance achieves better (or equivalent) optimum than the allocation with ex-ante relevance.*

*Proof.* To see why this holds, let $\mathbf{z}^{(t)}$ be the optimal allocation vector for the segment auction with ex-ante relevance for $t$-th segment. Notice that with ex-post relevance, we can optimize the allocation vector $\mathbf{x}^{(t)} \in \Delta_n$ given the precomputed output $(y_i)_{i \in [n]}$, *i.e.,* $\mathbf{x}^{(t)} = \mathbf{x}^{(t)}(y_1, \ldots, y_n)$. Here, however, we can restrict $\mathbf{x}^{(t)}(y_1, \ldots, y_n) = \mathbf{x}^{(t)}$, *i.e.,* uniformly the same over the output, and let $\mathbf{x}^{(t)} = \mathbf{z}^{(t)}$. Then, the resulting allocation becomes exactly equivalent to the optimal allocation for the ex-ante relevance. Therefore, marginalizing over the output would yield the same value of the objective function, implying that segment auction with ex-post relevance can at least achieve such quantity. $\square$

# B   Complexity of mechanisms

One another important aspect of the segment auction is to display the finalized output as fast as possible to not deteriorate user experience. We here formally characterize the computational complexity required to run each mechanism from the user experience perspective. To simplify the statements, we restrict our focus to the single allocation setting, however, a similar argument carries over to the multi-allocation setting. Overall, to start generating $t$-th segment, the segment auction requires to retrieve relevance measures $q_i^{(t)}$ for every $i \in [n]$, decides the winner of the auction and corresponding payment. Note, however, that the payment does not need to be calculated immediately, so we will only consider the computational/query complexity that will be required until the LLM starts generating the output. For instance, one can store the data occurred during the segment auction and compute the payment in an asynchronous manner by looking up the historical data.

To quantify the overall latency of each mechanism, we define the notions of query complexity to each modules we have defined. First, the *LLM query* complexity denotes how many times the segment auction is required to call the *LLM oracle* which gives an output of desired length given a query. Next, the *relevance query* complexity denotes the number of times the segment auction calls the *relevance oracle* that computes the relevance between two given text documents. The relevance oracle here

---

[10]We here focus on the single-allocation setting for the ease of exposition, but the same argument carries over to the multi-allocation setting.

[11]We remark that this is impossible in [9], since the output is endogenous to the mechanism in their setup so that the mechanism computes the prominence of each ad in the output. On the other hand, since the generation of output is fully governed by LLM to optimize the output, our mechanism can retrieve the output as parameters as well.

can be deemed as an abstraction of the retrieval component in the RAG framework. Typically, it is expected that the LLM query would be much more expensive than relevance query, which will be again much expensive than the bitwise operation dealt in the standard time complexity.

The complexity of an auction will be characterized by query complexities to each oracle as well as a time complexity required throughout the computation of the mechanism.

**Theorem B.1.** *The single-allocation segment auction with ex-ante relevance has LLM query complexity of $O(1)$, relevance query complexity of $O(n)$, and time complexity of $O(n)$ to generate each segment.*

*Proof.* To execute the mechanism, it first needs to compute the relevance measure $q_i^{(t)}$ for each $i \in [n]$, which requires $n$ calls of relevance oracle. After then, we need to sample the random noise, and compute the largest and the second-largest perturbed bid $q_i^{(t)} b_i e^{\varepsilon_i}$, which requires $O(n)$ time complexity. Then, the output can be generated by a single query to LLM. This completes the proof. □

On the other hand, the following theorem explicitly shows that dealing with ex-post relevance requires more LLM query complexity.

**Theorem B.2.** *The single-allocation segment auction with ex-ante relevance has LLM query complexity of $O(n)$, relevance query complexity of $O(n)$, and time complexity of $O(n)$ to generate each segment.*

*Proof.* The only difference is that, to compute the relevance $q_i^{(t)}$ for each $i \in [n]$, it requires the computation of $y_i$ in advance, which requires $O(n)$ LLM query complexity. Note here that after the winner of the auction is selected, there is no need of further calling the LLM oracle since all the outputs are already generated. □

## C  Beyond the independent segment auction

In Section 3, we restrict our attention to the case in which the relevance measure does not depend on the previous segments generated, but only on the query. We here generalize this limitation by introducing a general segment auction. In the general segment auction, we allow the relevance to be a function of the previous segments, which implies that for $t \neq t' \in [T]$, it might be the case that $q_i^{(t)} \neq q_i^{(t')}$. Importantly, each $q_i^{(t)}$ is in fact a function of all the previous segments $y^{(1:t-1)}$ and decision variables $(i_1, i_2, \ldots, i_{t-1})$ therein. That is, the relevance should be indexed by $q_i^{(t),(i_1,\ldots,i_{t-1})}$. Thus, an allocation $(i_1, i_2, \ldots, i_T)$ that maximizes the logarithmic NSW should solve the following optimization problem.

$$\max_{(\mathbf{x}^{(1)},\ldots,\mathbf{x}^{(T)}) \in \Delta_n^T} \sum_{t \in [T]} \sum_{i \in [n]} v_i q_i^{(t),(i_1,\ldots,i_{t-1})} \log x_i^{(t)}.$$

Note here that once $x_i^{(t)}$ is determined, the subsequent segment $i_t$ will be sampled according to the proper probability distribution, which then affects the quantity $q_i^{(t),(i_1,\ldots,i_t)}$. Since each $q_i^{(t),(i_1,\ldots,i_{t-1})}$ can have arbitrary value, solving the optimization problem above requires (i) exhaustive search over the space $[n]^T$ to obtain the quantities $q_i^{(t),(i_1,\ldots,i_t)}$ for every $t \in [T]$ and every $(i_1, \ldots, i_T) \in [n]^T$, and (ii) optimization over each probability simplex $\mathbf{x}^{(t)}$ for $t \in [T]$ given the exponentially many parameters. Overall, this is computationally infeasible in practice, particularly given the limited latency in the online advertisement system.

Instead, our segment auction can be deemed as a greedy algorithm that approximates the globally optimal allocation rule. Indeed, given the previous tokens, the segment auction chooses the next token to maximize the logarithmic social welfare up to the subsequent round. This is straightforward to see since the logarithmic social welfare is decomposable over each token and our segment auction chooses a token with respect to the probability that maximizes the single round logarithmic social welfare for the next round.

**Proposition C.1.** *Given a query $x$ and previous segments $y^{(1:t-1)}$, the segment auction chooses the next token that maximizes LSW up to the next token.*

# D  Further notion of fairness and its connection to LSW

We here explain several standard fairness concepts widely used in the literature, and how they are relevant to LSW. Given $n$ agents, let $X \subseteq \mathbb{R}^n_{\geq 0}$ be the space of feasible allocation, $\mathbf{x} \in X$ be an allocation and $\mathbf{w} \in \mathbb{R}^n_{\geq 0}$ be a corresponding weights. Such a weight may represent an entitlement of an agent that is exogenous from the allocation rule. For example, in a network bandwidth allocation scenario, a user with a higher weight might represent a critical application that requires more guaranteed resources, and in economic planning, different sectors (e.g., healthcare, education) might be assigned different weights based on their societal importance, influencing the distribution of resources. Suppose that each agent $i \in [n]$ has a valuation function $v : X \to \mathbb{R}_{\geq 0}$.

The standard notion of Nash social welfare is defined as the following [30].

**Definition D.1** (Nash social welfare)**.** The weighted Nash social welfare is defined as $\prod_{i \in [n]} ((u_i(x_i))^{w_i})^{1/\sum_{i \in [n]} w_i}$. If $w_i = 1$ for $i \in [n]$, then it is simply called the Nash social welfare.

Nash social welfare recently receive a tons of attraction from the computer science as well as economics literature [6], since it well balances the trade-off between the efficiency and fairness.

The following notion of proportional fairness is broadly studied in the resource allocation literature, in particular from the network utility maximization perspective [20, 22, 46]

**Definition D.2** (Weighted proportional fairness)**.** The allocation $\mathbf{x} \in X$ satisfies weighted proportional fairness if for any other allocation $y \in X$, it satisfies

$$\sum_{i \in [n]} w_i \frac{y_i - x_i}{x_i} \leq 0.$$

If $w_i = w$ for $i \in [n]$, then this is simply called the proportional fairness.

In essence, if we want to increase a certain coordinate $i$'s allocated resource by $\delta$, then it comes at the cost of decreasing some other agent $j$'s resource by $\delta$ (or summing over others), and proportional fairness precisely implies that the overall change of proportional utility $\delta/x_i - \delta/x_j$ is always nonpositive.

Finally, these notions have close connection to the following competitive equilibrium from equal incomes [42, 2, 5].

**Definition D.3** (CEEI)**.** Consider $n$ agents and $m$ (divisible) goods. Each good $j \in [m]$ has a supply of $s_j$. Each agent has a budget of $w_i$ unit of currency. Let $p_j$ be the price of good $j$. Let $u_i(\mathbf{x}_i)$ be the utility functoin of agent $i$ for the bundle $\mathbf{x}_i = (x_{i1}, x_{i2}, \ldots, x_{im})$ where $x_{ij}$ represents the quantities of each good $j$ allocated to agent $i$. A competitive equilibrium is a set of prices $\mathbf{p}$ and allocation $\mathbf{x}$ such that (i) each agent maximizes their utility, *i.e.*, $x_i \in \arg\max_y u_i(y)$ subject to $\sum_{j \in [m]} p_j y_j \leq 1$, and (ii) the market clears, *i.e.*, $\sum_{i \in [n]} x_{ij} = s_j$ for $j \in [m]$.

These concepts are independently discovered by several different communities including economics as the solution of the bargaining problem [30] and as the concept of competitive equilibrium [42], and also as the solution network scheduling problem [20]. It was a folklore knowledge for decades that all these notions induce the same allocation vector for divisible goods.

**Theorem D.4.** *Given weights* $\mathbf{w}$, *an allocation* $x \in X$ *that maximizes the weighted NSW satisfies weighted proportional fairness, and coincides with the competitive equilibrium given the weighted budget.*

Interestingly, our notion of logarithmic social welfare can be deemed as a version of the weighted NSW in which $q_i v_i$ is the weight $w_i$, and the utility is only about the allocation $x_i$, since LSW = $\prod_{i \in [n]} x_i^{q_i v_i}$. The main difference is that the actual utility is decomposed into two parts of allocation $(x_i)$ and the per-allocation utility $(q_i v_i)$, and further the monetary transfer is not considered at all. Therefore, one might interpret LSW as a measure that captures the balance between the *allocational* efficiency and fairness given that each advertiser has an entitlement of $q_i v_i$. Its further connections to proportional fairness and CEEI can be analogously argued as per Theorem D.4,

# E Combinatorial segment auction

We here present a combinatorial generalization of the multi-allocation segment auction presented in Section 3.2. Let $\mathcal{A} = 2^{[n]}$ be the power set over $[n]$, and $\mathcal{A}_k = \{A \in \mathcal{A} : |A| = k\}$. In the RAG-sequence model [23] as written in (1), the term $p_\eta(z_i|x)p_\theta(y|x, z_i)$ corresponds to conditioning the output $y$ on each single document $z_i$.[12] Recall that $\mathcal{A}_k$ is the collection of set of ads whose cardinality is $k$. To advertise $k$ ads within a single segment, we introduce the following equation similar to (2) using a combinatorial variant of RAG-sequence model.

$$P(y^{(t)}|x, y^{(1:t-1)}) = \sum_{A \in \mathcal{A}_k} P_\eta(z_A|x, y^{(1:t-1)}; \mathbf{b}) P_\theta(y^{(t)}|x, z_A, y^{(1:t-1)}), \qquad (6)$$

where $z_A$ is a document that represents the ads included in $A$.

To construct our intuition towards the auction, first assume that the bids are uniformly the same. Then, our each probability in the RAG equation should boil down to $P_\eta(z_A|x, y^{(1:t-1)})$. Here, similar to Assumption 2.1, we can deem this probability as an indirect measure of the relevance of the set of ads $A$ to the query $x$. However, we cannot exactly relate the average CTR of individual ad $i \in A$ with $P_\eta(z_A|x, y^{(1:t-1)})$, since $z_A$ only accounts for the set-wise property but not how much individual ad $i$ is relevant to the query context.

Alternatively, we can decompose the probability $p_\eta(z_A|x, y^{(1:t-1)})$ by a summation over the *prominence* of each ad $i \in A$ that contributes to the overall relevance of $A$, *i.e.,*

$$P_\phi(z_A|x, y^{(1:t-1)}) = \sum_{i \in [n]} P_\phi(z_i|x, y^{(1:t-1)}, z_A) P_\phi(z_A|x, y^{(1:t-1)}),$$

where $\sum_{i \in [n]} P_\phi(z_i|x, y^{(1:t-1)}) = 1$. The parameter $\phi$ denotes the model that computes the prominence of each $i$ and overall relevance $A$, which is different from the previous retriever's parameter $\eta$ since it will be calculated in a different manner. Note here that we specify the relevance of the set $A$ to be retrieved from a different model with $\phi$ not $\eta$, since if we simply append or summarize a set of ad documents and measure the relevance, there could be some biases due to positional effects. This implies one may need to implement an individual module to capture the overall relevance. For instance, one can use a heuristic of $q_A = \alpha \cdot \sum_{i \in A} q_i + \beta \sum_{i \neq j \in A} \text{rel}(a_i, a_j)$ for a proper choice of weights $\alpha \geq 0$ and $\beta \geq 0$. We also emphasize that our mechanism does not control the prominence but is given by the LLM in a exogenous manner, so one only needs to implement a proper relevance/prominence computation module that is well-calibrated with the CTR.

Then, we can impose the following assumption analogous to Assumption 2.1.

**Assumption E.1.** In a $t$-th segment, let $C_{A,i}^{(t)}$ be the binary event that ad $i$ is clicked in $t$-th segment if it is advertised with the set of ads $A$. We assume that

$$q_{A,i}^{(t)} \equiv \mathbf{E}[C_{A,i}^{(t)}|x, y^{(1:t-1)}] = \int_{y_A^{(t)}} \mathbf{E}[C_{A,i}|x, y^{(1:t-1)}, y_A^{(t)}] dy_A^{(t)} \propto p_\phi(z_i|x, y^{(1:t-1)}, z_A) p_\eta(z_A|x, y^{(1:t-1)}).$$

Correspondingly, we can define the linear aggregation function as follows.

$$\hat{q}_A^{(t)} = p_\phi(z_A|x, y^{(1:t-1)}; \mathbf{b}) = \frac{\sum_{i \in A} q_{A,i}^{(t)} b_i}{\sum_{B \in \mathcal{A}_k} \sum_{i \in B} q_{B,i}^{(t)} b_i}. \qquad (7)$$

If all the bids are the same, this reduces to the baseline RAG equation (6) such that $p_\eta(z_A|x, y^{(1:t-1)}) = \sum_{i \in A} p_\phi(z_i|x, y^{(1:t-1)}, z_A) p_\eta(z_A|x, y^{(1:t-1)})$, which is proportional to $\sum_{i \in A} q_{A,i}$.

Finally, given the number of winners $k$, the combinatorial segment auction for $t$-th segment can rigorously defined as follows.

---

[12]Note that even though we compute the normalized probability by summing over $i \in [n]$, this is not about conditioning the output on multiple ads, but rather conditioning the output on each single ad, and selecting the output by marginalizing over every possible choice of the single ads.

- Collect $\mathbf{b}$ and $\mathbf{q}^{(t)} = (q_{A,i}^{(t)})_{A \in \mathcal{A}_k, i \in A}$
- Draw $\varepsilon_A \sim \text{Gumbel}(0, 1)$ for each $A \in \mathcal{A}_k$.
- Compute $s_A = \sum_{i \in A} q_{A,i}^{(t)} b_i \cdot e^{\varepsilon_A}$.
- Pick $A^* = \text{argmax}_{A \in \mathcal{A}_k} s_A$.
- For each $i \in A^*$:
  - Find $A'(i) = \text{argmax}_{A \in \mathcal{A}_k : i \notin A} s_A$.
  - Charge each $i \in A^*$ the following VCG price per click:

$$p_i = \left( s_{A'(i)} - \left( \sum_{j \in A^* \setminus \{i\}} q_{A^*,j}^{(t)} b_j e^{\varepsilon_{A^*}} \right) \right) / (q_{A^*,i}^{(t)} e^{\varepsilon_{A^*}}).$$

We further assume an analogue of Assumption 3.1 stating that the advertiser's utility is additive over each segment, *i.e.,* per-impression utility is $u_i = \sum_{t \in [T]} u_i^{(t)} = \sum_{t \in [T]} v_i q_{A,i} x_i^{(t)} - p_i^{(t)}$.

Unlike the single allocation setting, ad $i$'s utility is positive whenever a set $A$ that includes $i$ is selected. Thus, we define the following variant of the weighted NSW.

**Definition E.2.** The combinatorial logarithmic social welfare (CLSW) is defined as follows.

$$\text{CLSW} = \prod_{i \in [n]} \left( \prod_{A \in \mathcal{A}_k : i \in A} x_A^{q_{A,i}^{(t)}} \right)^{v_i}.$$

The following theorem states that the presented multi-allocation segment exhibits several nice properties, proof of which can be found in Appendix H.

**Theorem E.3.** *Given a query $x$ and number of slots $k$, the combinatorial segment auction is DSIC, IR, Pareto efficient and has the maximal CLSW among independent segment auctions.*

We further show that it exhibits the following complexity measures.

**Proposition E.4.** *Combinatorial segment auction has LLM query complexity of $O(1)$, relevance query complexity of $O(kn^k)$, and time complexity of $O(n^k)$.*

*Proof.* The proof directly follows from the fact that it first needs to compute the set-wise segment for every $A$ and correspondingly the individual relevance for each set, which requires $\binom{n}{k} \times k$ query to the relevance measure. Then, computing the perturbed score and selecting the winner requires the computational complexity of $O(n^k)$. The LLM generation can be done by a single call once the mechanism finishes. $\square$

One issue of the combinatorial segment auction is that it does not always induce nonnegative payment, unlike the standard VCG payment. In the VCG payment, the payment is always guaranteed to be nonnegative due to the monotonicity of the social welfare. To see this more formally, let $W$ be the optimal social welfare with optimal set $A^*$, and let $Vi_i$ be $i$'s utility contributed for $i \in A^*$. Let $W_{-i}$ be the optimal social welfare when the agent $i$ is excluded from the society. Then, VCG charges the externality to each $i \in A^*$, *i.e.,* $p_i = W_{-i} - \sum_{j \neq i, j \in A^*} V_j$. Here, $p_i$ is guaranteed to be nonnegative because computing $W_{-i}$ includes the choice of $A^* \setminus \{i\}$. Thus, $W_{-i}$ is always larger or equivalent to $\sum_{j \neq i, j \in A^*} V_j$.

In the combinatorial segment auction, however, our choice over the $i$-excluded optimal social welfare $W_{-i}$ does not subsume the choice of $A^* \setminus \{i\}$ since we restrict our choice to be within the set with cardinality $k$. Hence, there might be a chance that $p_i$ is often negative. Indeed, letting $B = A^* \setminus \{i\}$, if $q_{B,j}$ for $j \in B$ is significantly smaller $q_{A^*,j}$ for $j \in A^*$, then it might happen that the social welfare of $B$ with cardinality $k - 1$ is much smaller than that of $A^*$ with $i$ added. In this case, since adding $i$ to the $B$ increases the participant's utilities, it is reasonable to *compensate* the agent $i$ with negative payment.

One way to prevent such a phenomenon is to simply round up the negative payment to be zero once it occurs. On the other hand, our perturbation based implementation of the segment auction enforces

that the random noise $\varepsilon_i$ is multiplied as an exponent do $e^{\varepsilon_i}$. This often incur a large gap between the first and the second highest score in the segment auction, inducing a higher chance of resulting negative payment. To prevent this, one may consider truncate the Gumbel distribution to be drawn from, or controlling the temperature parameter in the discrete choice model [41], both of which comes at the cost of having a slightly different allocation function.

# F LLM, prompts, and further experimental details

We use publicly-available API access of GPT4 as our LLM in our experiments. In this Section, we first provide the set of prompts used to generate output in our experiments. Then in F.1 we discuss our communication protocol with GPT4 API, and in F.2, we provide more details on experimental setup, including Naive II mechanism and ads used in our experiments.

We use the following prompts to generate output:[13]

**Single-allocation segment auction.** We start with `init_query` and continue the conversation with the model with `rest_query`. Note that `ad` refers to the `advertiser` context – a document describing its products or services.

```
prompt = '"Can you suggest some books similar to \"To Kill a Mockingbird\"?"'

init_query = f'''
        {prompt}\n please respond to this question for only one
        sentence while also advertise {advertiser} with this context >>
        {ad}
        Make sure to connect the answer and the advertisement very naturally,
        not something like appending the ads after just answering the question.
        Focus on answering the question,
        there shouldn't be too much advertisment in the output.
        Make sure that response should be one sentence.
        '''

rest_query = f'''
        You must continue your answer to my original query.
        Your previous response was
        >> {previous_output}

        And you now should advertise {advertiser},
        but without hurting the coherency of the entire document.
        Here's some contexts about {advertiser}

        >>  {ad}

        Make sure that there is one new sentence.
        Write the entire document, which merges your previous response and new paragraph.
        '''
```

**Multi-allocation auction.** We mainly did out experiments with three ads, but this template could be further generalized for more/fewer number of ads.

```
prompt = '"Can you suggest some books similar to \"To Kill a Mockingbird\"?"'

query = f'''{prompt}\n please respond to this question for only three sentence while
        (1) advertise {advertisers[0]} with this context >>
        {ads[0]}
```

---

[13]We remark that our prompt engineering might not be optimal, so better/different prompt engineering might result in better outputs.

```
(2) advertise {advertisers[1]} with this context >>
{ads[1]}

(3) advertise {advertisers[2]} with this context >>
{ads[2]}

Make sure to connect the answer and the advertisement very naturally,
not something like appending the ads after just answering the question.
Focus on answering the question,
there shouldn't be too much advertisment in the output.
Make sure to advertise all three brands and
ensure that the response is three sentences.
'''
```

### F.1 Configuration of prompt the LLM

Here we provide our protocol of communication with GPT4 model. `messages` refers to the history of chat between the model and us (client).

```
response = client.chat.completions.create(
        model = "gpt-4-turbo",
        logprobs = False,
        temperature = 1,
        max_tokens = 300,
        messages=messages,)
```

### F.2 Further experimental details

The following is a detailed pseudocode of Naive II mechanism.

---

**Naive II mechanism**

1. Collect $\mathbf{q}$ and $\mathbf{b}$.
2. Draw $\varepsilon_i \sim \text{Gumbel}(0, 1)$ for each $i \in [n]$ independently.
3. Compute the score $s_i = b_i e^{\varepsilon_i}$.
4. Select the winner $i^* = \text{argmax}_{i \in [n]} s_i$.
5. Find the second highest $i' = \text{argmax}_{i \in [n] \setminus \{i^*\}} s_i$.
6. Find the smallest bid $z$ for $i^*$ such that $s_{i^*} \geq s_{i'}$, which is $z = b_{i'} e^{\varepsilon_{i'}} / e^{\varepsilon_{i^*}}$.
7. Charge $z$ to $i^*$ *per click*.

---

Figure 5: Naive II mechanism

The following are one-sentence description of the ad listed in Table 5, followed by the ad document which is actually used in the prompt to incorporate each ad in the output generation process. Each ad is a mocked version of a real-world company named by LLM (guess what?), and all the relevant texts are generated by LLM as well.

1. Velora: A tech company that designs and sells premium, seamlessly integrated smart devices and services for a sophisticated and efficient lifestyle.

> Discover the future of technology with Velora, the brand that redefines innovation and elegance. Velora designs and sells a premium range of smartphones, tablets, laptops, and smartwatches, all crafted to seamlessly integrate into your lifestyle. Our products are engineered with user-friendly interfaces, stunning designs, and cutting-edge technology to keep you connected and productive. Velora's ecosystem offers unparalleled synchronization across devices, ensuring a smooth and efficient experience whether you're at work, school, or on the go. With Velora Pay, you can enjoy secure and convenient payment services, while our robust cloud service keeps your data safe and accessible anytime, anywhere. Elevate your tech experience with Velora, where sophistication meets simplicity and advanced functionality.

2. [BookHaven](): An online bookstore offering a vast selection of books across all genres with a seamless shopping experience and reliable delivery.

> Introducing BookHaven, your ultimate online bookstore where the world of literature is just a click away. At BookHaven, we offer an extensive collection of books spanning every genre and interest, from timeless classics and gripping thrillers to insightful non-fiction and enchanting children's stories. Our user-friendly platform ensures a seamless shopping experience, with personalized recommendations and unbeatable prices. Whether you're a voracious reader or just looking for your next great read, BookHaven is dedicated to delivering literary treasures right to your doorstep with fast, reliable shipping and a hassle-free return policy. Discover the joy of reading with BookHaven, where every book finds its perfect reader. Dive into a world of endless possibilities and let your next adventure begin at BookHaven!

3. [MassMart](): A membership-based retail store offering premium bulk products at unbeatable prices with a focus on customer satisfaction and community support.

> Experience the joy of shopping at MassMart, where quality meets value in a dynamic retail environment tailored for your satisfaction. At MassMart, members enjoy exclusive access to a vast selection of premium, bulk-sized products, from fresh groceries to high-tech electronics, all at unbeatably low prices. With a commitment to customer happiness, sustainability, and community support, MassMart isn't just a shopping destination — it's a part of your community. Dive into a world of savings and discover why millions choose MassMart as their trusted shopping partner. Join us today and see the difference MassMart can make in your shopping experience, where every visit is more than just shopping — it's an adventure!

4. [EspressoEdge](): A premium coffee shop offering high-quality, handcrafted beverages made from the finest Arabica beans, providing a luxurious coffee experience for all.

> Experience the warmth and delight of EspressoEdge, where every sip offers an invitation to a world of exquisite flavors and aromas. Renowned globally for its high-quality, handcrafted beverages, EspressoEdge is committed to sourcing the finest Arabica beans, expertly blending them into a variety of rich espressos, frothy cappuccinos, and creamy lattes. Each visit to an EspressoEdge store is more than just a coffee run—it's an opportunity to savor a moment of luxury amid the hustle of daily life. Whether you seek the comfort of a familiar classic or the thrill of a new seasonal specialty, EspressoEdge welcomes all to gather, connect, and enjoy a cup perfectly tailored to your taste. Step into your local EspressoEdge today and join us in celebrating the art of coffee.

5. [SocialHub](): A leading social media platform that connects over two billion users through personalized news feeds, interactive groups, and tools for sharing life's moments and promoting businesses.

> Discover the power of connection with SocialHub, the world's leading social media platform. With over two billion active users, SocialHub is your gateway to staying in touch with friends and family, discovering new communities, and sharing your life's moments. Our innovative features, from personalized news feeds to interactive groups, make it easy to engage with what matters most to you. Whether you're promoting your business, staying updated on the latest news, or simply keeping up with loved ones, SocialHub is the ultimate tool to enhance your digital experience. Join us today and be part of a global network where connections come to life!

6. [ColaBubbles](): The world's favorite soft drink, known for its unique flavor blend and effervescent bubbles that have been delighting people for over a century.

> Experience the refreshing taste of ColaBubbles, the world's favorite soft drink. With its unique blend of flavors and effervescent bubbles, ColaBubbles has been bringing joy to people of all ages for over a century. Whether you're enjoying a moment of relaxation, celebrating with friends, or on the go, ColaBubbles is the perfect companion to quench your thirst and uplift your spirits. Our commitment to quality and tradition ensures every sip is as delightful as the first. Indulge in the classic taste of ColaBubbles and make every moment special. Taste the feeling!

7. [FizzyPop](): An iconic soft drink celebrated for its crisp, refreshing flavor and vibrant effervescence, perfect for those who live life boldly and seek excitement in every moment.

> Unleash the bold taste of FizzyPop, the iconic soft drink that invigorates and refreshes like no other. Known for its crisp, refreshing flavor and vibrant effervescence, FizzyPop is the perfect choice for those who dare to live life to the fullest. Whether you're at a party, watching a game, or simply taking a break, FizzyPop brings a burst of excitement to any occasion. With a heritage of quality and a commitment to innovation, every sip of FizzyPop delivers an unmatched experience. Embrace the bold, and make every moment extraordinary with the unmistakable taste of FizzyPop.

8. [SkyTech](): The world's leading aerospace company, designing and manufacturing advanced commercial airplanes, defense systems, and space technologies to ensure safe and efficient global connectivity and exploration.

> Explore the skies with SkyTech, the world's leading aerospace company renowned for its innovation, quality, and reliability. SkyTech designs, manufactures, and services commercial airplanes, defense systems, and space technologies, making global connectivity and exploration possible. Whether you're traveling for business or leisure, SkyTech's state-of-the-art aircraft ensure a safe, comfortable, and efficient journey. With a legacy of pioneering advancements and a commitment to excellence, SkyTech continues to shape the future of aviation. Choose SkyTech and experience the pinnacle of aerospace engineering and performance. Fly with confidence, fly with SkyTech.

9. [AeroDynamics](): The global leader in aerospace innovation, designing and manufacturing advanced commercial aircraft that provide unparalleled comfort, efficiency, and reliability for a superior flying experience.

> Experience the future of aviation with AeroDynamics, the global leader in aerospace innovation and excellence. AeroDynamics designs and manufactures the world's most advanced commercial aircraft, providing unparalleled comfort, efficiency, and reliability. From cutting-edge technology to sustainable solutions, AeroDynamics is dedicated to shaping the future of air travel. Whether you're embarking on a long-haul journey or a short domestic flight, AeroDynamics ensures a superior flying experience with spacious cabins, innovative features, and top-notch safety standards. Trust AeroDynamics for a seamless and enjoyable journey every time. Fly smarter, fly with AeroDynamics.

10. MusicStream: The ultimate destination for streaming millions of songs with personalized recommendations and offline listening capabilities, offering a seamless music experience anytime, anywhere.

> Immerse yourself in the world of music with MusicStream, the ultimate destination for streaming your favorite tunes anytime, anywhere. With a vast library of millions of songs, playlists curated just for you, and personalized recommendations, MusicStream puts the power of music discovery in your hands. Whether you're in the mood for chart-topping hits, underground gems, or soothing melodies, MusicStream has something for everyone. Plus, with offline listening capabilities and seamless integration across devices, you can take your music with you wherever you go. Join the millions of music lovers worldwide and unlock endless possibilities with MusicStream. Discover, stream, and experience the joy of music like never before.

11. BrainChips: The global leader in semiconductor technology, providing cutting-edge processors that power a wide range of devices with industry-leading performance, reliability, and security for professionals, gamers, and more.

> Experience the cutting-edge innovation of BrainChips, the global leader in semiconductor technology. BrainChips' groundbreaking processors power the devices that fuel our modern world, from laptops and desktops to servers and cloud computing systems. With a legacy of pushing the boundaries of technology, BrainChips continues to deliver industry-leading performance, reliability, and security. Whether you're a professional tackling complex tasks or a gamer seeking immersive experiences, BrainChips processors provide the power and efficiency you need. Trust BrainChips to deliver the performance you demand and the reliability you can count on. Join the millions who rely on BrainChips technology and unlock new possibilities for productivity, creativity, and entertainment.

# G   Further experimental results

We here provide further experimental results that could not discussed in the main paper. For the auction outcomes, Scenario 1 denotes the setup presented in Section 4.

## G.1   Further results on the auction outcomes with different scenarios

| Advertiser | Bid | $q_i$ | $x_i$ | Mechanism | Soc. Wel. | Revenue | Relevance | Min. Soc. Wel. |
|---|---|---|---|---|---|---|---|---|
| Velora | 2 | 0.36 | 0.22 | Seg w/ repl. | .898 ($\pm$.0022) | .347 ($\pm$.0071) | .527 ($\pm$.0077) | .439 |
| BookHaven | 1 | 0.87 | 0.26 | Seg w/o repl. | .896 ($\pm$.0013) | .317 ($\pm$.0060) | .521 ($\pm$.0040) | .490 |
| MassMart | 3 | 0.31 | 0.28 | Naive II | .897 ($\pm$.0023) | .378 ($\pm$.0069) | .418 ($\pm$.0053) | .287 |
| EspressoEdge | 3 | 0.26 | 0.24 | Multi alloc | .892 ($\pm$.0013) | .255 ($\pm$.0058) | .516 ($\pm$.0042) | .515 |

Table 3: Setup of Scenario 2 representing an almost uniform allocative vector (left), and the corresponding auction outcomes (right).

**Scenario** 2**: Almost uniform allocation vector.** In this scenario, the allocation probabilities are almost the same across the four ads. The bids **b**, computed relevance **q**, and the allocation probability

**x** as well as the auction outcomes are presented in Table 3. Since all the advertisers induce almost the same allocative social welfare $q_i v_i$, one can verify that the overall social welfare does not significantly differ across the mechanisms in this case.

Since Naive II only accounts for the bids regardless of the relevance, we observe that the revenue is indeed the highest among three auctions. However, the overall relevance of the Naive II mechanism is much lower than both the segment auctions, which implies that the user experiment might be much worse than the segment auctions. For the minimum social welfare, Naive II exhibits significantly smaller quantity which is due to the nonuniform bids across the ads in Table 3.

| Mechanism | Soc. Wel. | Revenue | Relevance | Min. Soc. Wel. |
|---|---|---|---|---|
| Seg w/ repl. | 0.898 (±0.0022) | 0.347 (±0.0071) | 0.527 (±0.0077) | 0.439 |
| Seg w/o repl. | 0.896 (±0.0013) | 0.317 (±0.0060) | 0.521 (±0.0040) | 0.490 |
| Naive II | 0.897 (±0.0023) | 0.378 (±0.0069) | 0.418 (±0.0053) | 0.287 |
| Multi alloc | 0.892 (±0.0013) | 0.255 (±0.0058) | 0.516 (±0.0042) | 0.515 |

Table 4: Auction outcomes for Scenario 2.

**Scenario 3: More number of ads.** Here we consider 11 different advertisers as follows.

| Adv | Velora | Book Haven | Mass Mart | Espresso Edge | Social Hub | Cola Bubbles | Fizzy Pop | Sky Tech | Aero Dynamics | Music Stream | Brain Chips |
|---|---|---|---|---|---|---|---|---|---|---|---|
| Bids | 1 | 1 | 1 | 1 | 1 | 1 | 1 | 1 | 1 | 1 | 1 |
| $q_i$ | 0.36 | 0.87 | 0.31 | 0.26 | 0.21 | 0.36 | 0.38 | 0.28 | 0.33 | 0.34 | 0.33 |
| $x_i$ | 0.088 | 0.215 | 0.076 | 0.064 | 0.053 | 0.088 | 0.095 | 0.070 | 0.082 | 0.084 | 0.082 |

Table 5: Bids and relevance of the advertisers for Scenario 3.

Briefly speaking, an ad ('BookHaven') has relatively large allocation probability of 0.21 than any others, while all the others have similar allocation probability of around 0.08. We mainly observe a similar tendency discussed so far in the first two Scenarios.

| Mechanism | Soc. Wel. | Revenue | Relevance | Min. Soc. Wel. |
|---|---|---|---|---|
| Seg w/ repl. | 0.507 (±0.0068) | 0.482 (±0.0070) | 0.507 (±0.0068) | 0.039 |
| Seg w/o repl. | 0.489 (±0.0048) | 0.481 (±0.0074) | 0.489 (±0.0048) | 0.034 |
| Naive II | 0.423 (±0.0049) | 0.495 (±0.0071) | 0.423 (±0.0049) | 0.052 |
| Multi-alloc | 0.491 (±0.0049) | 0.453 (±0.0073) | 0.491 (±0.0049) | 0.042 |

Table 6: Auction outcomes for Scenario 3.

For the quality of the output, we again observe a similar tendency, however, the gap between the multi and single allocation segment auction amplifies.

| Mechanism | $k=1$ | $k=2$ | $k=3$ |
|---|---|---|---|
| Seg w/ repl. | 0.733 (±.005) | 0.701 (±.005) | 0.686 (±.004) |
| Seg w/o repl. | 0.745 (±.005) | 0.711 (±.004) | 0.697 (±.004) |
| Multi-alloc | - | - | 0.738 (±.004) |

Table 7: Similarity of output to the original output in Scenario 3.

## G.2 Further qualitative analysis

We here provide some qualitative analyses on the generated outputs from different mechanisms presented in Section 4. Relevant information about each ad can be found in the subsequent Appendix F.2. The following are examples of the original response with three sentences without any ad allocated.

> **Example 1.** If you enjoyed "To Kill a Mockingbird," consider reading "The Help" by Kathryn Stockett, which explores racial issues in 1960s Mississippi through the eyes of three women. Another excellent choice is "A Tree Grows in Brooklyn" by Betty Smith, a poignant coming-of-age story set in early 20th century Brooklyn. For a powerful narrative on racism and social justice, "The Secret Life of Bees" by Sue Monk Kidd is highly recommended, set in South Carolina during the 1960s civil rights movement.

> **Example 2.** If you liked "To Kill a Mockingbird," you might enjoy "The Help" by Kathryn Stockett, which delves into racial tensions in 1960s Mississippi. "A Tree Grows in Brooklyn" by Betty Smith offers a touching coming-of-age story in early 20th-century Brooklyn. Another great choice is "The Secret Life of Bees" by Sue Monk Kidd, set in 1960s South Carolina, exploring themes of racism and redemption.

> **Example 3.** For those who appreciated "To Kill a Mockingbird," "The Help" by Kathryn Stockett is a compelling choice, examining racial issues in 1960s Mississippi. "A Tree Grows in Brooklyn" by Betty Smith is another excellent option, featuring a heartfelt coming-of-age story in early 20th-century Brooklyn. Additionally, "The Secret Life of Bees" by Sue Monk Kidd, set in 1960s South Carolina, explores themes of racism and personal growth.

**With or without replacement.** In the setting of Scenario 2, the following is an output of the *segment auction with replacement*. Note that we explicitly specify the segment number for readability, although it was not directly specified in the LLM output.

> **(Segment 1)** At BookHaven, where every book finds its perfect reader, you'll discover amazing works similar to "To Kill a Mockingbird," such as Harper Lee's "Go Set a Watchman," Carson McCullers' "The Heart is a Lonely Hunter," and William Faulkner's "Intruder in the Dust," all ready to be explored and delivered right to your doorstep. **(Segment 2)** Dive into BookHaven, your ultimate online bookstore, to enjoy a user-friendly shopping experience with personalized recommendations and unbeatable prices, ensuring that every literary adventure is just a click away. **(Segment 3)** Plus, with our commitment to fast, reliable shipping and a hassle-free return policy, your next great read is guaranteed to arrive swiftly at your door.

In this random incidence of the segment auction with replacement, the same ad 'BookHaven' is repeatedly selected for all the segments. Notably, in the last segment LLM does not explicitly mention 'BookHaven', while keep talking about the benefits of 'BookHaven' actually (teal texts). From the user experience perspective, this would be better than keep mentioning an url to an ad for each segment, since the user may think the LLM is trying to advertise more but not to answer the query. We find this to be a potential advantage of allowing repeated selection of the same ad in the segment auction.

On the other hand, one might notice that the second segment is not coherently continued from the first segment's context (red texts). One might require a better prompt engineering, or allowing repetition might not be a good option for *coherency*, since it essentially requires several segments to repeatedly mention the same ad. This might be even more critical if the same ads are selected in a non-consecutive manner. One remedy might be to restrict the repeated selection to be allowable only if it is going to be consecutively selected in the neighboring segments.

The following is an output of the *segment auction without replacement*.

> **(Segment 1)** If you appreciated the profound narratives and moral questions explored in "To Kill a Mockingbird," you might enjoy "The Help" by Kathryn Stockett or "A Time to Kill" by John Grisham, both of which blend compelling storytelling with social issues, much like Velora blends sophistication and functionality in their cutting-edge devices. **(Segment 2)** Just as these books provide a backdrop to reflect on societal issues over a compelling story, visiting an EspressoEdge store can be your perfect escape to reflect and unwind with a meticulously crafted coffee, enhancing your experience of luxury and quality in every sip. **(Segment 3)** After indulging in coffee and social reflections, why not continue exploring similar profound narratives by visiting BookHaven, where a vast selection of literature awaits to complement your tastes and spark further thought, all conveniently available with just a click.

Interestingly, even though the LLM is forced to advertise different ad in each segment, we find that the resulting output is very coherent, in particular from how it begins the new sentence from the previous sentence's context (teal texts). Generally, however, when it allocates a less relevant ad like 'Velora', one may see that it is not very fluently advertised (red texts).

**Multi versus single allocation.**

Figure 6 is another pair of outputs for multi and single allocation segment auction.

> **Multi-allocation**:
> If you treasured "To Kill a Mockingbird," you might enjoy exploring similar themes of justice and morality in books like "The Help" by Kathryn Stockett or "A Time to Kill" by John Grisham, both of which you can easily find on BookHaven, where classic stories and modern narratives converge for readers like you. Between readings, you can dive into discussions with other book lovers or share your favorite quotes using SocialHub, transforming your reading experience into lively conversations and digital connections. Plus, imagine reading these great books on Velora's latest tablet, which offers not only an elegant design but also the cutting-edge technology to enhance your reading with features like blue light reduction for your comfort during long reading hours.
>
> **Single allocation**:
> (Segment1) If you appreciated the complex social themes and engaging narrative of "To Kill a Mockingbird," you might enjoy "The Secret Life of Bees" by Sue Monk Kidd, which explores similar questions of justice and identity in a setting that, much like Velora's seamlessly integrated technology, deeply connects and resonates with its environment. (Segment2) To easily find this book and many others that stir powerful discussions and thoughts, check out BookHaven, the ultimate online bookstore for a seamless shopping experience and personalized recommendations that ensure you find your perfect read, just as Velora ensures your tech experience is just right for your lifestyle. (Segment3) Furthermore, to discuss the impactful themes and characters found in both books, join global conversations on SocialHub, where you can connect with other literature enthusiasts and engage in meaningful discussions, bringing your reading experience into a vibrant community where every voice matters.

Figure 6: Outputs of multi and single allocation segment auction.

Again, we observe that the multi-allocation segment auction is superior in its quality from our (subjective) judgement. All the selected ads are very smoothly expanded throughout the response (teal texts). In the single allocation segment auction, however, the first segment includes 'Velora' in an awkward manner (red texts). Interestingly, the second segment in the single allocation segment auction once more mentions 'Velora' (red texts), even though only 'BookHaven' is selected in that segment. We believe this is because LLM tries to more naturally generate the second segment given the first segment, which in turn worsens its quality paradoxically.

**Longer segments.** Finally, we conduct an analogous experiment with longer segments (paragraph) using the single ad segment auction withour replacement. The following is an output of the *segment auction without replacement*.

> **(Segment 1)** If you appreciated the thematic depth and moral introspection of "To Kill a Mockingbird," you might enjoy exploring similar narratives like "The Help" by Kathryn Stockett or "A Time to Kill" by John Grisham. Find these titles easily at BookHaven, your ultimate online bookstore. With an extensive collection and personalized recommendations, BookHaven ensures a seamless shopping experience. Dive into a world of endless possibilities with BookHaven, where every book finds its perfect reader.
> **(Segment 2)** Additionally, consider delving into classics such as "Cry, the Beloved Country" by Alan Paton, which, like Harper Lee's masterpiece, offers profound insights into social justice and empathy. Pair your reading experience with a visit to EspressoEdge, where each sip of their high-quality, handcrafted beverages provides a moment of luxury. Savor rich espressos or creamy lattes while you immerse yourself in timeless literature at EspressoEdge.
> **(Segment 3)** For a contemporary twist, you might also enjoy "Small Great Things" by Jodi Picoult, a novel that tackles race and prejudice in modern society. Enhance your reading experience with Velora's range of tablets and e-readers, which offer crisp displays and user-friendly interfaces. Velora's smart devices ensure your favorite books are always accessible, whether you're at home or on the go. Elevate your tech experience with Velora.

Although not directly comparable with the previous result as the entire document itself gets longer, we find that the LLM keeps answering the original user's query in the beginning of each paragraph, and then try advertising the corresponding product more in a smooth manner (teal texts), whereas one might argue that there are some redundant texts advertising too much about the product (red texts), which might backfire for the marketing purpose.

## H Proofs

### H.1 Proof of Theorem 3.2

*Proof.* We first prove that the allocation vector induced by the segment auction is equivalent to (2). To this end, we use the following well known result from discrete choice model.

**Lemma H.1** (Chapter 3, [41]). *For each $i \in [n]$, let $s_i \geq 0$ be the score, and let $\tilde{s}_i$ be the perturbed score with i.i.d. random noise $\varepsilon_i$ drawn from Gumbel$(0, 1)$, i.e., $\tilde{s}_i = s_i + \varepsilon_i$. Then, the probability that $\tilde{s}_i$ has the largest value among $i \in [n]$ is $s_i/(\sum_{j \in [n]} s_j)$.*

Using the lemma above, it is straightforward to see that each ad $i$ is selected with probability $q_i b_i/(\sum_{j \in [n]} q_j b_j)$.

Now we show that such allocation vector maximizes the logarithmic social welfare, assuming the truthful bids, *i.e.,* $\mathbf{b} = \mathbf{v}$. Due to the independence, it suffices to show that the logarithmic social welfare for a fixed segment $t$ is maximized by our allocation rule. We take an inverse approach of finding the allocation that maximizes the LSW. That is, we are interested in $\mathbf{x}$ that maximizes

$$\text{NSW}^{(t)} = \prod_{i \in [n]} x_i^{q_i b_i}.$$

such that $\sum_{i \in [n]} x_i = 1$ and $x_i \geq 0$ for any $i \in [n]$. Set up the Lagrange function $L$ as follows.

$$L(\mathbf{x}, \lambda) = \prod_{i \in [n]} x_i^{q_i b_i} + \lambda(-1 + \sum_{i \in [n]} x_i).$$

By taking the partial derivative with respect to $x_i$ for $i \in [n]$ and $\lambda$, we obtain a system of equations

$$\frac{\partial L}{\partial x_i} = q_i b_i \prod_{i \in [n]} x_i^{q_i b_i}/x_i + \lambda = 0, \forall i \in [n]$$

$$\frac{\partial L}{\partial \lambda} = -1 + \sum_{i \in [n]} x_i = 0.$$

To solve for $\mathbf{x}$, we first obtain

$$q_i b_i \prod_{i \in [n]} x_i^{q_i b_i} / x_i = C,$$

for some constant $C$ for any $i \in [n]$. This yields proportional relationship of

$$\frac{q_i b_i}{x_i} = c,$$

for some constant $c$ for any $i \in [n]$. Plugging into $\sum_{i \in [n]} x_i = 1$, we get the desired allocation.

Finally, given any random noise $(\varepsilon_i)_{i \in [n]}$, we will show that the resulting realization of our segment auction is DSIC and IR. To prove IR, first consider the winner $i^*$. Its per-click utility is given by

$$u_{i^*} = v_{i^*} - \frac{q_j b_j e^{\varepsilon_j}}{q_{i^*} e^{\varepsilon_{i^*}}} \geq 0,$$

for second highest bidder index $j$ since $i^* = \text{argmax}_{i \in [n]} q_i b_i e^{\varepsilon_i}$. Since $u_j = 0$ for every other $j$, it is clear that IR holds. DSIC also naturally follows from the fact that the second price auction is DSIC, and the segment auction can be viewed as a randomization over the second price auction. To more explicitly prove this fact given the random noise, for the winner $i^*$, it is obvious that there exists no incentive to deviate due to the second price payment. Consider $j \neq i^*$, *i.e.*, $q_{i^*} b_{i^*} e^{\varepsilon_{i^*}} > q_j b_j e^{\varepsilon_j}$.[14] Originally $j$ realizes per-click utility of $u_j = 0$. Suppose $j$ increases its bid so that $b'_j$ satisfies $q_j b'_j e^{\varepsilon_j} \geq q_{i^*} b_{i^*} e^{\varepsilon_{i^*}}$. Then, $j$'s per-click utility will be

$$(u_j)' \leq v_j - \frac{q_{i^*} b_{i^*} e^{\varepsilon_{i^*}}}{q_j e^{\varepsilon_j}} < 0 < u_j.$$

Thus, $j$'s utility only decreases, and this concludes that the segment auction is DSIC for any random noise. Note that the same argument holds for per-impression utility. Finally, the Pareto-efficiency directly follows from the fact that the allocation is maximal. $\qquad\square$

## H.2 Proof of Theorem 3.3

*Proof.* To characterize the expected per-click payment formula, we use the lemma by [28].

$$p_i(\mathbf{q}, \mathbf{b}) = \int_0^{b_i} z \cdot \frac{d}{dz} \left( \frac{q_i z}{\sum_{j \in [n]} q_j b_j} \right) dz.$$

Write $w_{-i} = \sum_{j \in [n] \setminus \{i\}} q_j b_j$. Computing the integral above, we obtain

$$
\begin{aligned}
p_i(\mathbf{q}, \mathbf{b}) &= \int_0^{b_i} z \cdot \frac{d}{dz} \left( \frac{q_i z}{w_{-i} + q_i z} \right) dz \\
&= \int_0^{b_i} z \cdot \left( \frac{w_{-i}/q_i}{(w_{-i}/q_i + z)^2} \right) dz \\
&= \int_0^{b_i} \frac{w_{-i} z / q_i}{(w_{-i}/q_i + z)^2} dz \\
&= \frac{w_{-i}}{q_i} \left( \frac{w_{-i}/q_i}{z + w_{-i}/q_i} + \ln\left(z + w_{-i}/q_i\right) \right) \Big|_0^{b_i} \\
&= \frac{w_{-i}}{q_i} \left( \frac{w_{-i}}{w_{-i} + q_i b_i} + \ln(b_i + w_{-i}/q_i) - 1 - \ln(w_{-i}/q_i) \right) & (8) \\
&= \frac{w_{-i}}{q_i} \left( \ln(\frac{q_i b_i + w_{-i}}{w_{-i}}) - \frac{q_i b_i}{w_{-i} + q_i b_i} \right) & (9) \\
&\geq 0 & (10)
\end{aligned}
$$

[14]Note that we can ignore the tie-breaking case since Gumbel distribution is continuous, thereby $q_i b_i e^{\varepsilon_i}$ can be deemed as a sample from a continuous distribution for each $i \in [n]$.

where the last inequality follows from the fact $\ln(x)+1/x \geq 0$.[15] Individual rationality in expectation follows from the fact that each realized instance of segment auction is individual rational, however, to see this explicitly, observe that for each $i \in [n]$, per-click utility satisfies

$$
\begin{aligned}
u_i(\mathbf{q}, \mathbf{b}) &= b_i \frac{q_i b_i}{w_{-i} + q_i b_i} - \frac{w_{-i}}{q_i}\left( \ln(\frac{q_i b_i + w_{-i}}{w_{-i}}) - \frac{q_i b_i}{w_{-i} + q_i b_i} \right) \\
&\geq b_i \frac{q_i b_i}{w_{-i} + q_i b_i} - b_i + \frac{w_{-i} b_i}{w_{-i} + q_i b_i} \\
&= b_i \cdot 1 - b_i = 0
\end{aligned}
$$

where in the first inequality we use $-\ln(1+t) \geq -t$ for $t > -1$. $\qquad\square$

## H.3 Proof of Theorem E.3

*Proof.* We first prove that our allocation maximizes the CLSW. A similar Lagrangian-based argument implies that our allocation function maximizes the following over $x = (x_A)_{A \in \mathcal{A}_k}$ that belongs to $\binom{n}{k}$ dimensional probability simplex.

$$
(\hat{q}_A^{(t)})_{A \in \mathcal{A}_k} = \operatorname*{argmax}_{x \in \Delta_{\binom{n}{k}}} \sum_{A \in \mathcal{A}_k} \hat{q}_A^{(t)} \log x_A \tag{11}
$$

$$
= \operatorname*{argmax}_{x \in \Delta_{\binom{n}{k}}} \sum_{A \in \mathcal{A}_k} \left( \sum_{i \in A} q_{A,i}^{(t)} v_i \right) \log x_A \tag{12}
$$

$$
= \operatorname*{argmax}_{x \in \Delta_{\binom{n}{k}}} \sum_{i \in [n]} v_i \left( \sum_{A \in \mathcal{A}_k : i \in A} q_{A,i}^{(t)} \log x_A \right). \tag{13}
$$

Equivalently, this can be written as

$$
\prod_{i \in [n]} \left( \prod_{A \in \mathcal{A}_k : i \in A} x_A^{q_{A,i}^{(t)}} \right)^{v_i}, \tag{14}
$$

which implies that our allocation maximizes CLSW. DSIC, IR, and Pareto efficiency follow from the similar argument with Theorem 3.2, *i.e.,* from the fact that our payment function is VCG payment of charging the externality, which is equivalent to the Myerson payment in the single-dimensional setting. $\qquad\square$

## H.4 Proof of Theorem 3.3

*Proof.* To characterize the expected per-click payment formula, we use the lemma by [28].

$$
p_i(\mathbf{q}, \mathbf{b}) = \int_0^{b_i} z \cdot \frac{d}{dz}\left( \frac{q_i z}{\sum_{j \in [n]} q_j b_j} \right) dz.
$$

---
[15]We skip the elementary level algebraic manipulation.

Write $w_{-i} = \sum_{j \in [n] \setminus \{i\}} q_j b_j$. Computing the integral above, we obtain

$$
\begin{aligned}
p_i(\mathbf{q}, \mathbf{b}) &= \int_0^{b_i} z \cdot \frac{d}{dz} \left( \frac{q_i z}{w_{-i} + q_i z} \right) dz \\
&= \int_0^{b_i} z \cdot \left( \frac{w_{-i}/q_i}{(w_{-i}/q_i + z)^2} \right) dz \\
&= \int_0^{b_i} \frac{w_{-i} z / q_i}{(w_{-i}/q_i + z)^2} dz \\
&= \frac{w_{-i}}{q_i} \left( \frac{w_{-i}/q_i}{z + w_{-i}/q_i} + \ln(z + w_{-i}/q_i) \right) \Big|_0^{b_i} \\
&= \frac{w_{-i}}{q_i} \left( \frac{w_{-i}}{w_{-i} + q_i b_i} + \ln(b_i + w_{-i}/q_i) - 1 - \ln(w_{-i}/q_i) \right) && (15) \\
&= \frac{w_{-i}}{q_i} \left( \ln(\frac{q_i b_i + w_{-i}}{w_{-i}}) - \frac{q_i b_i}{w_{-i} + q_i b_i} \right) && (16) \\
&\geq 0 && (17)
\end{aligned}
$$

where the last inequality follows from the fact $\ln(x) + 1/x \geq 0$.[16] Individual rationality in expectation follows from the fact that each realized instance of segment auction is individual rational, however, to see this explicitly, observe that for each $i \in [n]$, per-click utility satisfies

$$
\begin{aligned}
u_i(\mathbf{q}, \mathbf{b}) &= b_i \frac{q_i b_i}{w_{-i} + q_i b_i} - \frac{w_{-i}}{q_i} \left( \ln(\frac{q_i b_i + w_{-i}}{w_{-i}}) - \frac{q_i b_i}{w_{-i} + q_i b_i} \right) \\
&\geq b_i \frac{q_i b_i}{w_{-i} + q_i b_i} - b_i + \frac{w_{-i} b_i}{w_{-i} + q_i b_i} \\
&= b_i \cdot 1 - b_i = 0
\end{aligned}
$$

where in the first inequality we use $-\ln(1 + t) \geq -t$ for $t > -1$. $\qquad\square$

## H.5   Proof of Theorem 3.4

For ease of exposition, we slightly redefine some notations. Let $N$ be the set of agents and $n$ be the number of agents. We overwrite $b_i = \log(q_i b_i)$ and therefore perturbation can be written as $b_i + \varepsilon_i$ instead of $q_i b_i e^{\varepsilon_i} = e^{\log(q_i b_i)} \cdot e^{\varepsilon_i}$. That is, each agent $i$ places bid $b_i$, which is perturbed by $\varepsilon_i \sim \text{Gumbel}(0, 1)$ to yield the final perturbed bid $B_i = b_i + \varepsilon_i$. We take the top $k$ ads according to perturbed bids, where $1 \leq k \leq n$. Recall that the pdf and cdf of a $\text{Gumbel}(0, 1)$ distribution are:

$$
\begin{aligned}
f(\varepsilon) &= e^{-\varepsilon} \cdot e^{-e^{-\varepsilon}} \\
F(\varepsilon) &= e^{-e^{-\varepsilon}}
\end{aligned}
$$

We want to evaluate the probability that the set of agents $S \subseteq N$ wins, where $|S| = k$. First we evaluate the cdf and pdf for the minimum perturbed bid among $S$.

$$
\begin{aligned}
\mathbb{P}(\min_{i \in S} B_i \geq u) &= \prod_{i \in S} (1 - F(u - b_i)) \\
&= \sum_{T \subseteq S} (-1)^{|T|} \prod_{j \in T} F(u - b_j) \\
&= \sum_{T \subseteq S} (-1)^{|T|} \exp \left( -\sum_{j \in T} e^{b_j - u} \right) \\
\mathbb{P}(\min_{i \in S} B_i \leq u) &= 1 - \sum_{T \subseteq S} (-1)^{|T|} \exp \left( -\sum_{j \in T} e^{b_j - u} \right)
\end{aligned}
$$

---

[16]We skip the elementary level algebraic manipulation.

Taking the derivative:

$$\mathbb{P}(\min_{i \in S} B_i = u) \;\; = \;\; \sum_{T \subseteq S} (-1)^{|T|+1} \exp\left(-\sum_{j \in T} e^{b_j - u}\right) \cdot \left(\sum_{j \in T} e^{b_j - u}\right)$$

The cdf for the maximum perturbed bid among $\bar{S}$ is as follows.

$$\mathbb{P}(\max_{i \in \bar{S}} B_i \le u) \;\; = \;\; \prod_{i \in \bar{S}} F(u - b_i)$$

$$= \;\; \exp\left(-\sum_{i \in \bar{S}} e^{b_i - u}\right)$$

Continuing, the probability that $S \subseteq N$ wins is as follows.

$$\mathbb{P}(S \text{ wins}) \;\; = \;\; \int_{-\infty}^{\infty} \mathbb{P}(\max_{i \in \bar{S}} B_i \le u) \cdot \mathbb{P}(\min_{i \in S} B_i = u)\, du$$

$$= \;\; \int_{-\infty}^{\infty} \exp\left(-\sum_{i \in \bar{S}} e^{b_i - u}\right) \cdot \sum_{T \subseteq S} (-1)^{|T|+1} \exp\left(-\sum_{j \in T} e^{b_j - u}\right) \cdot \left(\sum_{j \in T} e^{b_j - u}\right) du$$

$$= \;\; \sum_{T \subseteq S} (-1)^{|T|+1} \int_{-\infty}^{\infty} \exp\left(-\sum_{i \in \bar{S} \cup T} e^{b_i - u}\right) \cdot \left(\sum_{j \in T} e^{b_j - u}\right) du$$

$$= \;\; \sum_{T \subseteq S} (-1)^{|T|+1} \left(\sum_{j \in T} e^{b_j}\right) \int_{-\infty}^{\infty} \exp\left(-e^{-u} \sum_{i \in \bar{S} \cup T} e^{b_i}\right) \cdot e^{-u}\, du$$

We now do the change of variable $t = e^{-u}$, $dt = -e^{-u}\, du$. As $u \to -\infty$, $t \to \infty$, and as $u \to \infty$, $t \to 0$. Continuing:

$$\mathbb{P}(S \text{ wins}) \;\; = \;\; \sum_{T \subseteq S} (-1)^{|T|+1} \left(\sum_{j \in T} e^{b_j}\right) \int_{\infty}^{0} \exp\left(-t \sum_{i \in \bar{S} \cup T} e^{b_i}\right) (-dt)$$

$$= \;\; \sum_{T \subseteq S} (-1)^{|T|+1} \left(\sum_{j \in T} e^{b_j}\right) \int_{0}^{\infty} \exp\left(-t \sum_{i \in \bar{S} \cup T} e^{b_i}\right) dt$$

$$= \;\; \sum_{T \subseteq S} (-1)^{|T|+1} \left(\sum_{j \in T} e^{b_j}\right) \left(\frac{\exp\left(-t \sum_{i \in \bar{S} \cup T} e^{b_i}\right)}{-\sum_{i \in \bar{S} \cup T} e^{b_i}} \bigg|_{t=0}^{\infty}\right)$$

$$= \;\; \sum_{T \subseteq S} (-1)^{|T|+1} \left(\sum_{j \in T} e^{b_j}\right) \left(0 - \frac{1}{-\sum_{i \in \bar{S} \cup T} e^{b_i}}\right)$$

$$= \;\; \sum_{T \subseteq S} (-1)^{|T|+1} \frac{\sum_{j \in T} e^{b_j}}{\sum_{i \in \bar{S} \cup T} e^{b_i}}$$

To summarize, the probability that a set $S \subseteq N$ of size $1 \le k \le n$ wins is:

$$\mathbb{P}(S \text{ wins}) = \sum_{T \subseteq S} (-1)^{|T|+1} \frac{\sum_{j \in T} e^{b_j}}{\sum_{i \in \bar{S} \cup T} e^{b_i}}.$$

Plugging back $b_i = \log(q_i b_i)$ yields the desired allocation probability.

