# OpenReview forum: "Ad Auctions for LLMs via Retrieval Augmented Generation"
_NeurIPS.cc/2024/Conference — NeurIPS 2024 poster_

### Official Review · Reviewer_2L4f · 2024-06-17

**Soundness:** 2
**Presentation:** 2
**Contribution:** 1
**Rating:** 4
**Confidence:** 3

**Summary:**

The paper considers the integration of ads into large language model (LLM) generation, as well as the design of a mechanism for ad allocation and pricing that comes with this integration. The paper introduces the notion of a "segment auction", where the output discourse is break down into various segments. Following the retrieval augmented generation (RAG) framework, the auction selects winner for each segment considering the bids and relevance, and provide the winning ad to the LLM as part of the prompt to generate segment with the ad incorporated. The paper provides theoretical analysis on the desirable properties of the segment auction, as well as empirical analysis in a synthetic setting.

**Strengths:**

- The paper does a good job connecting the ideas to theoretical concepts, and provides solid theoretical analysis
- The paper is well-motivated, tackling a novel problem that has great practical implications

**Weaknesses:**

- The paper uses strong assumptions on the retrieval component being calibrated to the click through rate, which forms the basis of most theoretical results provided. In the experiment section, the relevance measure are instead estimated by a model from the sentence-transformers library, which leaves a gap between theoretical guarantees and empirical approaches.
- Under the general segment auction exposited in Appendix C, where each relevance measure is additionally dependent on all previous segments and allocations, the assumption of calibrated relevance seem even harder to achieve. Without this assumption and its associated theoretical properties, the paper presents only incremental technical contribution since running a second price auction and incorporating the winner of the auction into the LLM prompt lacks technical novelty.
- The experiments section in the paper is not compelling enough.
    - The five types of auctions considered in the Experiment section, which varies in whether replacement or relevance measure is used, are not well-motivated and appear disconnected from the earlier theoretical section, which focus on only the segment auction with replacement. The authors do not provide a compelling reasoning for considering this five variants, and the result analysis does not provide a strong case or intuition for which should be chosen (see line 273-287) - the writing and analysis for these parts could be improved.
   - Both the numbers in output quality and the qualitative example provided in Figure 4 suggests that the addition of ads in the generation quite notably impairs the quality of the generated content. Particularly worrisome is the fact that in both the single allocation and multi allocation example (as well as in the examples in Appendix G.2), as soon as the paragraph starts pivoting to the ads *in the first sentence*, it never goes back in providing *any* additional useful information to answering the original question *"Can you suggests books similar to `to kill a mocking bird'"*. This seems to empirically suggests that the presence of ads in the earlier segment may impair the quality of content generated in the following segments.
  - The experiments assume that each segment is one sentence in the experiments and add an ad to each segment. This is arguably too extreme for it to be deployed and a more realistic setting might be to integrate only a few ads to an entire page of results, which will suggests that the segment be longer than a single sentence. Additional experimental results should demonstrate that the proposed mechanism work as well under a longer segment.
- When the number and categories of the advertisers are not diverse enough, augmenting an irrelevant ad appears to come with a significant cost to the quality of the output. The bid system also introduces the potential problem of misleading and corrupting the accuracy of the response, which is a new problem not faced by traditional ad auctions. This should be either empirically evaluated or discussed in more detail (see more in Limitations part).
- Typo in line 140 - both the per-click and per-impression price are denoted using the same symbol.

**Questions:**

- How will the relevance measure $q_i$ be obtained if the mechanism is put into practice, and will there be ways of ensuring accuracy and calibration?
- The qualitative empirical results right now (e.g. Figure 4) shows that the generated output stops providing useful information to the user once the ad starts appearing in the generated text in the first sentence, and all subsequent generations are just rhetorical transitions to another ad. Is this effect prevalent? Are there additional qualitative or quantitative results showing the effects of ad on subsequent segments and potential solutions to preserve an informative output?
- Current pricing uses the VCG mechanism for incentive compatibility. In ad auctions GSP and first price auctions are frequently used. Can these alternative pricing schemes can be easily implemented under the current scheme?

**Limitations:**

The authors mentions the inherent trade-off between revenue and quality that's evident from the empirical observations. Different from traditional ad auctions, under LLM when the generation of later texts depends on prior generations, integrating additional ad leave open potential room for attack and injecting biased information through the type of the ad itself. The authors can provide a more detailed exposition of the potential social impact of such mechanism.

---

> ### Author Rebuttal · Authors · 2024-08-06
>
> We appreciate the reviewer for thoughtful and detailed comments.
>
> > The paper uses strong assumptions on the retrieval component
>
> We agree that implementing a performant, calibrated retrieval component is a challenging engineering task in practice. However, it is well studied in the literature. We refer to McMahan et al. [1] and Graepel et al. [2] for descriptions of actual click-through rate estimation systems at Google and Microsoft. Both papers have sections on calibration using isotonic regression. We will cite these works and add some more discussion of calibration to the paper based on the reviewer’s comments.
>
> For our experimental evaluation, we view retrieval and calibration as modular components, where various implementations could be used in practice. Our evaluation uses embedding similarity as a measure of similarity because RAG commonly uses vector embeddings for retrieval. Vector embeddings capture semantic information and enable efficient similarity search, but as mentioned above, other options are possible. As we did not conduct the real-world experiment with an implemented system, our experiments are bound to use a proxy measure. We also kindly refer to simultaneous work by Dubey et al. [3] that pose the same assumption on the calibrated relevance (in their words, prominence).
>
> 1. McMahan et al.: Ad Click Prediction: a View from the Trenches, KDD’13
> 2. Graepel et al.: Web-Scale Bayesian Click-Through Rate Prediction for Sponsored Search
> Advertising in Microsoft’s Bing Search Engine, ICML‘10
> 3. Dubey et al.: Auctions with LLM Summaries, Dubey, Feng, Kidambi, Mehta, Wang, KDD’24
>
> > Under the general segment auction exposited in Appendix C
>
> Modern click-through rate estimation and retrieval systems take many contextual features to make their predictions. We refer again to papers [1, 2] cited above. In our application, prior generated output would be included as contextual features. We agree this is important and nontrivial to handle in practice, but again view this aspect as orthogonal/modular to the question of auction design tackled in our paper.
>
>
> > The five types of auctions considered in the Experiment section
>
> Thank you for raising this point. In our camera-ready draft, we will elaborate on our choice of baselines and how our results highlight trade-offs between mechanisms in terms of output quality, revenue, and social welfare. Here’s a brief intuition. We refer to L#242-250 regarding different baselines and L#271-287 for interpreting the results.
>
> In summary: (1) Naive II generates the highest revenue but results in the lowest relevance and social welfare, potentially incorporating unrelated ads and diminishing user experience; (2) Segment auctions with replacement achieve the best social welfare and relevance but lead to lower minimum social welfare; (3) Multi-allocation produces competitive results with segment auctions without replacement but sacrifices revenue.
>
> Tables 2 and 7 evaluate LLM output under different mechanisms. We measure relevance to the original output (without ads). Multi-allocation mechanisms generally produce better outputs, as longer segments with more ads lead to more coherent results. Single-allocation auctions with shorter segments result in lower quality. Naive I and II lead to the lowest quality due to the inclusion of irrelevant ads. Segment auctions without replacement show marginal improvement over those allowing replacement.
>
> These results highlight the trade-offs between different criteria, guiding the auctioneer to select the appropriate mechanism or a combination based on their requirements.
>
>
> > Figure 4 suggests that the addition of ads impairs..
>
> We kindly refer the reviewer to global rebuttal. As seen in the attached pdf, in 3rd paragraph in the first sample and in particular 2nd paragraph in the second sample, the output still answers the query. These can further be improved with more prompt engineering.
>
> > Augmenting an irrelevant ad appears to come with a significant cost to the quality of the output
>
> Like standard search auctions, target ads can be prefiltered to exclude irrelevant ones. Most ad platforms use prefiltering to reduce computational load and maintain user satisfaction. Therefore, we assume all ads are relevant in L#31.
>
> > Privacy / reliability issue
>
> We agree that integrating ads without privacy or robustness issues is crucial. However, our main focus is to introduce the RAG integration into the LLM ad system, validate its feasibility, and derive theoretical and empirical evidence. Our paper doesn’t address every practical issue. Similar perspectives are taken by Duetting et al., Dubey et al., and Soumalias et al. For broader technical concerns like privacy and reliability in LLM ad systems, refer to Feizi et al.'s survey paper.
>
> 1. Dubey et al.: Auctions with LLM Summaries, KDD’24
> 2. Duetting et al.: Mechanism Design for Large Language Model, WWW’24
> 3. Soumalias et al.: Truthful Aggregation of LLMs with an Application to Online Advertising
> 4. Feizi et al: Online Advertisements with LLMs: Opportunities and Challenges
>
>
> > How will the relevance measure be obtained if the mechanism is put into practice?
>
> We refer to the cited papers on CTR prediction for calibration. While we use vector similarity as a proof of concept, more research is needed on designing effective metrics to measure relevance between queries and ads.
>
> > The qualitative results show that the generated output stops providing useful information.
>
> Please see our response above.
>
> > Can these alternative pricing schemes can be easily implemented?
>
> As can be seen from our randomized segment auction, any pricing scheme can be implemented from our idea - randomly perturb $q_i \cdot b_i$, and apply any mechanism as desired. For instance, perturbing the bids and then run GSP (Figure 3) does not induce incentive-compatible mechanism as VCG does, however, this would be a reasonable analogue of the GSP auction widely adopted in the sponsored search auction.

---

> > ### Comment · Reviewer_2L4f · 2024-08-07
> > **Response**
> >
> > I thank the authors for their detailed response.
> >
> > - I'm still not convinced with novelty.
> >   - The main theoretical innovation is to connect the RAG framework with ad auctions. This is achieved in the paper by incorporating bids into the generation probability (Eq. 3) through Gumbel perturbations, as the authors highlighted in the global rebuttal. I agree with the authors that this is a nice incorporation of a method used in econometrics. However, it appears lacking if the use of this trick is the main theoretical contribution of the paper.
> >   - The new notion of logarithmic social welfare is interesting. However, as the authors mentioned, it is a new concept has never been studied in the literature so I would be more convinced of its importance if there's more evidence supporting its merits and properties theoretically and/or empirically. Other theoretical derivations like proof for DSIC and the pricing rule bear close semblance to the ad auctions literature. Overall, I find the research question interesting, but still not finding a lot of theoretical innovations in the proposed approach.
> > - I'll be more convinced of the merits and practical importance of the approach if there's strong empirical evidence. I thank the authors for the additional pdf they uploaded in the global rebuttal. I'm still not fully convinced that it resolved my point in the review:
> > > Particularly worrisome is the fact that in both the single allocation and multi allocation example (as well as in the examples in Appendix G.2), as soon as the paragraph starts pivoting to the ads in the first sentence, it never goes back in providing any additional useful information to answering the original question "Can you suggests books similar to `to kill a mocking bird'".
> >   - The authors pointed to the "3rd paragraph in the first sample and in particular 2nd paragraph in the second sample". I don't really see it in the 3rd paragraph in the first sample, maybe I'm missing something? For the 2nd paragraph in the second sample, I do see it: *"Additionally, consider delving into classics such as "Cry, the Beloved Country" by Alan Paton, which, like Harper Lee's masterpiece, offers profound insights into social justice and empathy"*. This seems good. Although in this case *both* the first and the second paragraph are advertising **MassMart**, so it seems to be a case where LLM doesn't need to make transitions. Are there examples where the paragraphs are advertising different companies but the subsequent paragraphs still add meaningful materials to the question? Would also help if we could see how much prompt engineering will be needed to achieve that and whether it's ad hoc to a single question. Ideally we want a prompt structure that works for at least a subset of questions.

---

> ### Author Response · Authors · 2024-08-08
>
> We appreciate the reviewer's responsiveness.
>
> ### Novelty
>
> We first emphasize that our main contribution is to propose the idea of combining RAG and ad auctions for LLM and laying down a foundation in this line of research. Indeed, even though applying RAG into the LLM ad auction seems a very plausible direction, there has been no single work dedicated to it. That is to say, our primary objective and contribution is to propose a potentially promising framework for the LLM ad auctions, which we believe will inevitably emerge in a near future, and validate its feasibility.
> Although RAG was introduced in 2020 and advertising has long been a fascinating field of study, the integration of ads within the RAG framework (to potentially monetize LLMs) is a novel approach that was not studied theoretically nor validated through empirical research.
>
> As the reviewer agreed with the idea of perturbation, although this idea is easily understandable in hindsight, it is not easy to come up with at first glance. Also, we think that the idea of reverse-engineering the mechanism design by starting from an exogenously given allocation function by RAG is not a trivial perspective to derive. We respectfully disagree with the reviewer that a paper necessarily includes complicated theoretical techniques or proofs, but believe that simple and concise but novel ideas often advance the fields in many directions.
>
> Finally, as the reviewer pointed out, delving more into logarithmic social welfare seems very promising research directions. We indeed have exposited several initial properties in Appendix D including that it can be viewed as a version of weighted Nash social welfare, but further theoretical results remains open problems, e.g., whether this is consistent with other notions of fairness in terms of reachability/approximability. We will make this point clearer in the main body of the paper.
>
>
> ### Experiments with longer segments
>
> We would like to emphasize that optimizing the incorporation of ads into the LLM output, via prompt engineering, is not a goal of this paper. Our intention in the experiments was to provide a proof of concept with basic, straightforward prompts, to confirm that the approach was viable (and surely in practice we may apply this approach for longer answers etc for quality purposes).
>
>
> Having said that below we bring another sample that we obtained using the single ad segment auction without replacement, even without much of prompt engineering (we just reused the original prompts but slightly added a bit: keep answering the query without deviating from it by fully focusing on the ads). We conducted this experiment several times, and found that LLM has been pretty consistent on almost all of them, i.e., a rather simple prompting structure would work.
>
> > (Segment 1) If you appreciated the thematic depth and moral introspection of "To Kill a Mockingbird," you might enjoy exploring similar narratives like "The Help" by Kathryn Stockett or "A Time to Kill" by John Grisham. Find these titles easily at BookHaven, your ultimate online bookstore. With an extensive collection and personalized recommendations, BookHaven ensures a seamless shopping experience. Dive into a world of endless possibilities with BookHaven, where every book finds its perfect reader.
>
> > (Segment 2) Additionally, consider delving into classics such as "Cry, the Beloved Country" by Alan Paton, which, like Harper Lee's masterpiece, offers profound insights into social justice and empathy. Pair your reading experience with a visit to EspressoEdge, where each sip of their high-quality, handcrafted beverages provides a moment of luxury. Savor rich espressos or creamy lattes while you immerse yourself in timeless literature at EspressoEdge.
>
> > (Segment 3) For a contemporary twist, you might also enjoy "Small Great Things" by Jodi Picoult, a novel that tackles race and prejudice in modern society. Enhance your reading experience with Velora's range of tablets and e-readers, which offer crisp displays and user-friendly interfaces. Velora's smart devices ensure your favorite books are always accessible, whether you're at home or on the go. Elevate your tech experience with Velora.
>
>
> We can definitely include these samples in the appendix for the camera ready if the reviewer finds it informative.

---

> ### Comment · Reviewer_2L4f · 2024-08-09
>
> Thanks for the detailed response and the new example, this is helpful to know. I'll think about the authors response, and wait for the other reviewers' response and discussion before making a final decision on my rating.
>
> Edit: The new example given by the authors partly addresses a concern raised in my original review so I will slightly adjust my rating accordingly. My other points remain the same at the moment.

---

### Official Review · Reviewer_xTCf · 2024-07-11

**Soundness:** 3
**Presentation:** 3
**Contribution:** 3
**Rating:** 7
**Confidence:** 4

**Summary:**

This paper studies an interesting and timely application of ad auctions for LLMs via retrieval augmented generation. They propose a segment auction that takes the bid and relevance as the input and outputs the price by a randomized second price auction. This auction maximizes the logarithmic social welfare that is proposed in this paper.

**Strengths:**

- The combination of RAG and Ad Auction is pretty interesting and novel.
- The presentation of the figures is clear and informative.
- This paper has empirical experiments.

**Weaknesses:**

- The underlying assumption of this paper, i.e., [line 188-189] the relevance is independent of the previous segment is too strong.

**Questions:**

In the paper, does the auction repeat for $T$ times, where one for each segment?

**Limitations:**

Yes.

---

> ### Author Rebuttal · Authors · 2024-08-06
>
> We appreciate the reviewer’s valuable comments.
>
> > Independent assumption
>
> We in fact extend our independent segment auction to a more general segment auction with history dependent relevance measure in Appendix C. We will make sure to add a pointer to this part in the main body of the paper. In the dependent segment auction, our notion of relevance depends on the previous tokens. In this case, since the relevance measure that determines the value of the objective function LSW might arbitrarily depend on the previous tokens, from the computational perspective, it requires the extension of every possible token sequence to exactly optimize the objective function without no assumption. Interestingly, we show that  our segment auction can be viewed as a local greedy algorithm that approximates the globally optimal allocation rule. More precisely, our segment auction chooses next tokens to maximize LSW given that previous tokens are fixed.
>
>
> > Question on the repetition
>
> The reviewer is correct - our segment auction repeatedly runs to determine the output for each segment, where there are flexibilities in determining what to choose as a segment. Also our multi-ad segment auction further allows flexibility in determining how many ads to allocate within a segment.

---

> > ### Comment · Reviewer_xTCf · 2024-08-07
> >
> > Thanks for the rebuttal. After I reviewed all the reviews/rebuttals, I believe this paper is an important and timely starting point for pricing RAG-based LLM, so I raised my score accordingly.

---

> > > ### Author Response · Authors · 2024-08-09
> > >
> > > We appreciate your positive feedback. Thanks a lot for raising the score!

---

> ### Comment · Reviewer_xTCf · 2024-08-13
>
> After reviewing the authors' rebuttal and other reviews, I have several points to raise:
>
> **Modeling Approach**: The paper employs hard insertion instead of model fusion to recommend an RAG-based sponsored search. This choice seems suboptimal, especially considering concurrent theoretical work (see https://arxiv.org/pdf/2407.04471) that utilizes specific model fusion methods for similar tasks. However, given that this concurrent work was published post NeurIPS 2024 deadline, it's understandable that the authors did not reference it.
>
> **Empirical Performance**: I concur with Reviewer 2L4f's concerns regarding empirical performance. More comprehensive evaluations are needed to substantiate the proposed method's efficacy.
>
> I still maintain my current score, but I want to second that there are indeed some points worth noticing in this paper.

---

### Official Review · Reviewer_VuMT · 2024-07-13

**Soundness:** 2
**Presentation:** 2
**Contribution:** 2
**Rating:** 4
**Confidence:** 4

**Summary:**

In this paper, the authors integrate the auction mechanism into RAG LLMs for computational advertising. They propose a novel segment auction method where an auction is run to integrate single or multiple ads into each segment output of LLMs. Experiments on several auction scenarios are conducted to verify the effectiveness and feasibility of the proposed framework.

**Strengths:**

1.	The research problem of this paper is interesting. This paper combines the popular RAG LLM framework with traditional auction mechanisms, exploring the prospects of integrating computational advertising with LLMs.
2.	This paper is well-structured with a clear and coherent logic.

**Weaknesses:**

1.	The technical novelty of the proposed method seems limited.
2.	The experimental evaluation should include more baseline methods.
3.	The evaluation method is not comprehensive enough.

Detailed Comments:
1.	The proposed method lacks innovation. While combining computational advertising with LLMs is an interesting direction, this paper merely provides a simplistic integration of auction mechanisms and RAG, lacking innovation in its overall approach.
2.	In the evaluation of the proposed method, this paper only compares two naive baselines (without relevance score / without an LLM), lacking comparison with other existing auction methods. The comparison should be made within the same RAG framework, between the proposed auction mechanism and other existing auction mechanisms, to demonstrate that the proposed mechanism is most compatible with RAG LLM.
3.	The effectiveness of the whole proposed framework is not well verified. First, simply comparing the cosine similarity of embeddings between the output text and the original text without ads may not sufficiently indicate the text quality. On one hand, comprehensive metrics like perplexity could be incorporated. On the other hand, the quality of the advertising content in the text has not been considered. Second, there is a lack of overall results that can demonstrate how well this method achieves the tradeoff between advertising effectiveness, output quality, allocating efficiency, and fairness.

**Questions:**

Refer to detailed comments.

**Limitations:**

Refer to detailed comments.

---

> ### Author Rebuttal · Authors · 2024-08-06
>
> We appreciate the reviewer for detailed review. We hope that our rebuttal can address your concerns.
> Note that our framework consists of several components, each of which could be further researched to improve real-world deployment. Our focus was on proposing the framework, analyzing it theoretically, and evaluating it with simple yet efficient methods.
>
> > The technical novelty of the proposed method seems limited.
>
> We kindly ask the reviewer to see our general response in “Novelty of the paper”
>
> > The experimental evaluation should include more baseline methods.
>
> Our framework on ad auctions LLM is indeed the first study to explore ad auction framework within the LLM’s textual output, so we compare our mechanisms with simple mechanisms that one might naively think of. There are simultaneous works by Aranyak and Soumalias, however, we remark that these works do not explicitly compare their mechanisms with one another due to the lack of existing/standard benchmarks, but also only considers a naive version of auctions, e.g., greedy append of ad documents similar to our naive mechanism. We kindly ask the reviewer to let us know if the reviewer has any specific auction to compare in mind.
>
> > First, simply comparing the cosine similarity of embeddings between the output text and the original text without ads may not sufficiently indicate the text quality.
>
> Measuring text quality while including advertisements is a challenging research question that requires further exploration by the NLP community. Traditional metrics like ROUGE may be unsuitable due to ads in the output and the absence of ground truth. While our framework's validation employed semantic similarity as a proxy for quality, qualitative analysis suggests our framework generally produces high-quality outputs. We acknowledge that this aspect could open a promising research area.
>
> > On one hand, comprehensive metrics like perplexity could be incorporated.
>
> Thank you for your valuable suggestion. We found that comparing perplexity is indeed useful and conducted experiments using the LLaMA3 8B model. We measured perplexity (given the query) for outputs in scenario 1 (100 samples per mechanism). Perplexity was low for original outputs but increased significantly when ads were included. Auction mechanisms that allow replacements showed slightly lower perplexity due to ad replication, which reduces model surprise. Other baselines performed competitively in terms of perplexity.
>
> | mechanism                        | perplexity |
> |----------------------------------|------------|
> | original                         | 4.93  |
> | single-alloc w replacement    | 12.66 |
> | multi-alloc greedy               | 14.69 |
> | single-alloc w/o replacement | 14.44 |
>
> Note that while perplexity is a useful measure, it doesn't fully capture how well the model responds to the query or the overall output quality. This limitation highlights an interesting problem that requires further research to develop better metrics for evaluating response relevance and quality.
>
>
> > On the other hand, the quality of the advertising content in the text has not been considered.
>
> Thanks to the reviewer for pointing this out, It indeed raises an interesting and fundamental research question for the general RAG framework: how effectively does the model utilize retrieved documents? There has been extensive study to evaluate RAG output given its retrieved documents. We believe that this problem requires further analysis in ad domain specifically.
>
> In our framework, we assumed for simplicity that when an ad is selected, the LLM can effectively integrate it into the output. Qualitative results support this assumption, demonstrating that the model properly blends ads into the generated output.
>
>
>
> > Second, there is a lack of overall results that can demonstrate how well this method achieves the tradeoff between advertising effectiveness, output quality, allocating efficiency, and fairness.
>
> We note that qualitative results show that our proposed approach is feasible and recent LLMs like GPT4 are useful for this framework. We used simple but efficient proxies for measuring ads relevance and output quality that compares different baselines and show that output quality of our mechanism is pretty reasonable. A bit more formally, advertising effectiveness and allocating efficiency can be captured via social welfare, output quality can be captured via relevance, and fairness can be captured via minimum social welfare as stated in L#260~265.

---

> > ### Author Response · Authors · 2024-08-10
> >
> > We thank you again for your valuable feedback and comments which have helped to strengthen our paper. As the discussion period is ending soon, we would really appreciate if you could let us know if our responses have addressed your concerns. We will be happy to answer any further questions and address any remaining concerns to the best of our abilities in the remaining time!

---

> > > ### Author Response · Authors · 2024-08-13
> > >
> > > Dear reviewer, As the discussion period is ending soon, we would really appreciate if you could let us know if our detailed responses have addressed your concerns and that possibly you can upgrade your score given other reviews with high score for this paper. Thanks again for your valuable comments and time.

---

### Official Review · Reviewer_zmfd · 2024-07-15

**Soundness:** 3
**Presentation:** 3
**Contribution:** 4
**Rating:** 7
**Confidence:** 4

**Summary:**

The work studies ad auctions integrated in LLM output powered by retrieval-augmented generation. The authors propose an ad auction (so-called segment auction) where an ad is put in retriever with some probability. An efficiency-fairness balance is maximized (through logarithmic social welfare). An extension to multi-ad setup is proposed. Evaluation of the framework has been done through synthetic experiments.

**Strengths:**

-	Highly important novel topic for research (due to restructuring of web search market and LLM/RAG-based search system deployment around the world)

-	Both theoretical guarantees and empirical evaluation

**Weaknesses:**

-	Unclear selection of optimization function (see questions)

-	It would be nice to have real-life evaluation as the topic is highly relevant for production search engines

**Questions:**

-	It is not very clear why does LSW has been selected for optimization? Is it chosen after segment auction invention? Or you purposely searched for auction that optimize LSW?

-	Does segment auction maximize SW? What is the auction design for SW optimality?

**Limitations:**

the authors adequately addressed the limitations

---

> ### Author Rebuttal · Authors · 2024-08-06
>
> We first appreciate the reviewer for insightful comments.
>
> > About unclear selection of optimization function
>
> The reviewer is correct that it is chosen after the allocation function of segment auction is determined. We remark that our selection of optimization functions stems from the probabilistic retrieval nature of the RAG framework. RAG framework suggests retrieving each document proportional to the probability $p_\eta(a_i | x)$. Thus, we restrict our mechanism to have the same allocation function so that the overall architecture exactly resembles the RAG framework but with ad documents. In hindsight, we observe that such an allocation function maximizes LSW, which has never been studied in the literature to the best of our knowledge. We firmly believe that LSW may play a crucial role as an objective function in ad auctions for LLMs in further works, and also we found that it achieves a balance between fairness and efficiency in a similar vein with the Nash social welfare.
>
> > Does segment auction maximize SW? What is the auction design for SW optimality?
>
> In fact, an (single-allocation) auction that maximizes SW will be any auction that selects an ad with the highest value of $q_i \cdot b_i$, e.g., second-price auction or first-price auction. As per the revenue-equivalence theorem by Myerson’82, all these auctions would induce the same revenue, whereas our segment auction intentionally deviate from such SW-maximizing allocation function to be consistent with the probabilistic retrieval of RAG, i.e., in order to improve the overall quality of LLM outputs themselves.

---

### Author Rebuttal · Authors · 2024-08-06

We thank the reviewers for their constructive feedback. In this section, we address the concerns regarding design choices and engineering components in our proposed framework, along with a recap over our main contributions, and further results for multi-ad segment auction.

------

### Segments and Prompt Engineering

+ choice of segments: the segment size is intentionally set and is not constrained by our mechanism or theoretical results. It's determined based on our understanding of the large language model's (LLM) ability to incorporate ads within each segment's output, in order to focus on delivering our main takeaway. In the paper, we used smaller segments to produce shorter, more efficient outputs and provide clearer qualitative results. The success of the LLM with shorter segments suggests that our approach will also work for longer segments. We have included some qualitative examples for **longer segments (e.g., paragraphs) in the PDF file**.

+ output quality: The quality of the outputs can be improved by refining the prompts given to the language model, ensuring better integration of ads within the output while effectively responding to users’ queries. Effective prompt engineering can lead to more coherent outputs that meet the needs of both advertisers and users.

While we appreciate the reviewers' insights on these choices, we emphasize that the main contribution of our paper is to propose and theoretically analyze this framework, alongside empirical evaluation. There is always room for improvement in each component by selecting better options.

------

### Novelty of the paper

We believe that the main technical novelty lies in applying the RAG framework into the ad auction for LLM and revealing its several properties.

+ First, our main mechanism considers a completely different approach than the standard mechanism design literature since we consider the allocation function as given from the RAG’s probabilistic retrieval nature. Then, given the allocation function as fixed, we aim to discover which objective such an allocation function tries to achieve, which turns out to be the logarithmic social welfare we defined. We firmly believe this approach could be a stepping stone to explore the intersection of RAG and mechanism design.

+ In addition, our perturbation technique inspired by discrete choice methods is indeed a novel component to implement any such randomized allocation rule in a simplistic manner. For instance, if one wants to naively implement a linearly proportional allocation rule, the straightforward approach would be to simply randomize the allocation function and sample from the corresponding function. In this approach, however, the payment function is not easy to obtain. One may apply Myerson’s lemma to obtain the expected payment given the randomized allocation function, however, this only guarantees incentive-compatibility in an ex-ante manner. In stark contrast, our perturbation technique ensures that the segment auction is ex-post incentive-compatible if the mechanism given the perturbed bids is incentive-compatible. Even more interestingly, in our framework, one can apply arbitrary pricing schemes given by any auction literature, e.g., generalized second price auction (GSP), even though it is not incentive-compatible. The approach is again to apply random perturbations to the bids, and then use the GSP pricing rule with the resulting fixed bids. Note that this is not possible without perturbation technique, e.g., it is not straightforward to answer what is the analogue of GSP when the allocation function needs to be randomized.
We remark that there has been no work to consider the idea of RAG + LLM ad auction and investigate the underlying theoretical / empirical nature of the corresponding framework in detail.

+ Finally, we evaluated various segment auction mechanisms and other ad auction baselines within the RAG framework. Our analysis provides insights into auction outcomes, such as social welfare and revenue, in some realistic scenarios, providing practical guidance for implementing this framework in real-world use cases.

------

### Further result on the multi-ad segment auction

We actually have obtained the allocation function of the GSP analogue under the RAG framework. Precisely, for the multi-ad segment auction presented in Figure 3, we observe that the allocation probability can be computed as follows:
$$ P(S \text{ wins }) = \sum_{T \subseteq S} (-1)^{|T|+1} \frac{\sum_{j \in T} q_jb_j}{\sum_{i \in \bar{S}\cup T}q_ib_i},$$
which strictly generalizes the single allocation segment auction by taking $S$ to be a set including $i$. We will add a brief remark on this result in the camera-ready.

---

### Decision · Program_Chairs · 2024-09-25

**Decision:**

Accept (poster)

**Comment:**

While there was a general consensus in the reviews that this paper is providing a plausible initial approach to an important emerging domain, the discussion focused on two aspects.  First, given the experimental results is the approach promising enough.  Second, is the market design sufficiently novel and interesting.

The general consensus is that the experiment results are relatively weak.  The authors stressed in their response that these results are not the main focus of the paper, and I and some others agree that in an early paper like this it isn't essential for the experimental results to be state of the art or even really practical.

This makes the second question more important, and after the discussion there remained at least some disagreement.  My own option, shared with some of the reviewers, is that even if the design doesn't have much technical novelty in terms of introducing major new building blocks or analysis techniques, it does combine existing techniques in a novel way to create a design that is able to achieve desirable properties in a novel and interesting setting. (Arguably, the limited technical novelty is even a benefit in terms of being able to explain the way the market works to participants.)